# Social experience shapes fighting strategies in *Drosophila*

**Can Gao[1†], Mingze Ma[1†], Jie Chen[1], Xiaoxiao Ji[1], Qionglin Peng[1]\*, Yufeng Pan[1,2]\***

[1]The Key Laboratory of Developmental Genes and Human Disease, Jiangsu Key Laboratory of Brain Science and Medicine, School of Life Science and Technology, Southeast University, Nanjing, China; [2]Co-innovation Center of Neuroregeneration, Nantong University, Nantong, China

**\*For correspondence:**
pengqionglin@seu.edu.cn (QP);
pany@seu.edu.cn (YP)

[†]These authors contributed equally to this work

## eLife Assessment

The **important** paper presents a new behavioral assay for *Drosophila* aggression and demonstrates that social experience influences fighting strategies, with group-housed males favoring high-intensity but low-frequency tussling over aggressive lunging observed in isolated males. The experiments are **solid** and the conclusions are of interest to researchers studying the impact of social isolation on aggression.

## Abstract

Social isolation generally increases aggression but decreases mating competition, resulting in an intricate and ambiguous relationship between social experience, aggression, and reproductive success. In male *Drosophila*, aggression is often characterized by lunging, a frequent and comparatively low-intensity combat behavior. Here, we provide a behavioral paradigm for studying a less frequent but more vigorous fighting form known as tussling. While social enrichment decreases lunging, aligning with past observations, it heightens the more forceful tussling behavior. These two forms of aggression rely on different olfactory receptor neurons, specifically Or67d for lunging and Or47b for tussling. We further identify three pairs of central pC1 neurons that specifically promote tussling. Moreover, shifting from lunging to tussling in socially enriched males is accompanied by better territory control and mating success. Our findings identify distinct sensory and central neurons for two fighting forms and suggest that social experience may shape fighting strategies to optimize reproductive success.

## Introduction

Competition for resources such as territory, food, and mates is crucial for animal survival and reproduction (*Bergman et al., 2010*; *Hoffmann and Cacoyianni, 1990*; *Stockley and Campbell, 2013*; *Zwarts et al., 2012*). Among diverse competitive strategies, physical combat is a prevalent form of fighting to repel rivals and regulated by both genetic factors and social environment (*Chen and Sokolowski, 2022*; *Zwarts et al., 2012*). Understanding how social experience influences combat behaviors in animals to maximize their survival and reproduction remains a fundamentally important yet inadequately understood question.

In the fruit fly, *Drosophila melanogaster*, inter-male aggression includes relative low-intensity fights such as lunging, in which one fly rears up on its hind legs and snaps down onto its opponent, as well as relative high-intensity fights like tussling, where two males tumble over each other (*Certel and Kravitz, 2012*; *Chen et al., 2002*; *Hoopfer, 2016*; *Palavicino-Maggio and Sengupta, 2022*). A recent work showed that lunges are insufficient to establish dominance but mainly serve to maintain social status, while tussling frequency is correlated with dominance establishment (*Simon and Heberlein, 2020*).

It is noteworthy that the majority of studies focus on the low-intensity lunging behavior, owing to its higher frequency and ease of observation. Substantial studies have identified regulatory genes and the neural circuit underlying the lunging-based aggressive behaviors (*Hoopfer, 2016*; *Kravitz and de la Fernandez, 2015*). In particular, the male-specific volatile pheromone 11-*cis*-vaccenyl acetate (cVA) acutely promotes lunging behavior via its olfactory receptor Or67d (*Wang and Anderson, 2010*). Interestingly, chronic cVA exposure in socially enriched males reduces the level of lunging via another olfactory receptor Or65a (*Liu et al., 2011*). Consistently, it has been found that social isolation increases lunging frequency (*Palavicino-Maggio and Sengupta, 2022*; *Yadav et al., 2024*) by altering the expression of genes such as *Cyp6a20* (*Wang et al., 2008*), *Drosulfakinin* (*Dsk*) (*Agrawal et al., 2020*), and *Tachykinin* (*Tk*) (*Asahina et al., 2014*). Social isolation also increases male aggression in mice (*Matsumoto et al., 2005*; *Zelikowsky et al., 2018*) and even in humans (*Ferguson et al., 2005*). In mice, the TK homologue Tachykinin 2 (Tac2) responds to social isolation and promotes inter-male aggressive behaviors (*Zelikowsky et al., 2018*). These findings contribute to the consensus that social isolation escalates aggression, whereas social enrichment mitigates it in animal models.

Meanwhile, social enrichment increases male mating success by enhancing the sensitivity of *Or47b*-expressing olfactory receptor neurons (ORNs) to fly pheromones common to both sexes (*Lin et al., 2016*; *Sethi et al., 2019*). Artificially activating *Or47b* ORNs also improves mating advantage in a competitive assay (*Dweck et al., 2015*). Group housing and age-dependent juvenile hormone (JH) jointly increase the expression of the male-specific proteins from *fruitless* (*fru*M), which is a master gene for male courtship behaviors (*Peng et al., 2021*; *Sato et al., 2019*; *Yamamoto and Koganezawa, 2013*), in the *Or47b*-expressing ORNs (*Hueston et al., 2016*; *Sethi et al., 2019*; *Zhao et al., 2020*). Notably, courtship levels in single-housed (SH) wild-type males are equal to *Dankert et al., 2009*; *Pan and Baker, 2014*, in some cases, even stronger than those in group-housed (GH) males (*Dankert et al., 2009*). In fact, group housing decreases excitability of the courtship-promoting P1 neurons (*Inagaki et al., 2014*), which express both terminal genes in the sex determination hierarchy, *fru* and *doublesex* (*dsx*). These findings raise a question on how social enrichment, which suppresses lunging-based aggression and does not increase courtship intensity, would facilitate mating competition.

Here, we provide a behavioral paradigm for studying inter-male tussling in *Drosophila* and discover that this infrequent yet intense form of combat is enhanced by social enrichment, while the low-intensity lunging is suppressed by social enrichment. This social enhancement of tussling requires olfactory receptor Or47b, but not Or67d or Or65a. We further identify three pairs of central *dsx*-expressing pC1 neurons that specifically promote tussling. Moreover, we demonstrate that socially enriched males can overcome mating disadvantages associated with aging.

## Results

### A paradigm for high-intensity aggression in *Drosophila*

While studying the lunging-based aggression in *Drosophila* using a high-throughput aggression assay based on previously aggression paradigms (*Certel and Kravitz, 2012*; *Wu et al., 2020*; *Zhou et al., 2008*), we occasionally observed the high-intensity boxing and tussling behavior in male flies as previously reported (*Chen et al., 2002*; *Nilsen et al., 2004*), which are difficult to distinguish and hereafter simply referred to as tussling. We reasoned that the intensity, instead of the frequency, could be decisive for a combat (*Simon and Heberlein, 2020*). Thus, although tussling is a less frequent behavior, it could serve a vital function.

To induce a higher level of tussling for further manipulation, we first introduced a fixed live virgin female in the same behavioral chamber for assaying lunging. The female fly was immobilized in the middle of the food, with its head embedded in the food and its body exposed to enhance male competition (*Figure 1A*). We found that males who were group-housed for 7 days showed a significantly higher frequency of tussling behavior and reduced latency to tussle in the presence of an embedded virgin female (*Figure 1B*). We next compared tussling in male flies of different ages. We observed a significantly higher proportion of 14-day-old males raised in groups that showed tussling behavior during the 2 hr observation period, compared to the 7-day-old males (*Figure 1B*). Furthermore, 2-day-old males never displayed tussling behavior. In contrast, 14-day-old males showed a lower level of lunging behavior than 7-day-old males did (*Figure 1—figure supplement 1*). Previous studies have shown that, in addition to mating resources, food quality is another crucial factor affecting aggressive

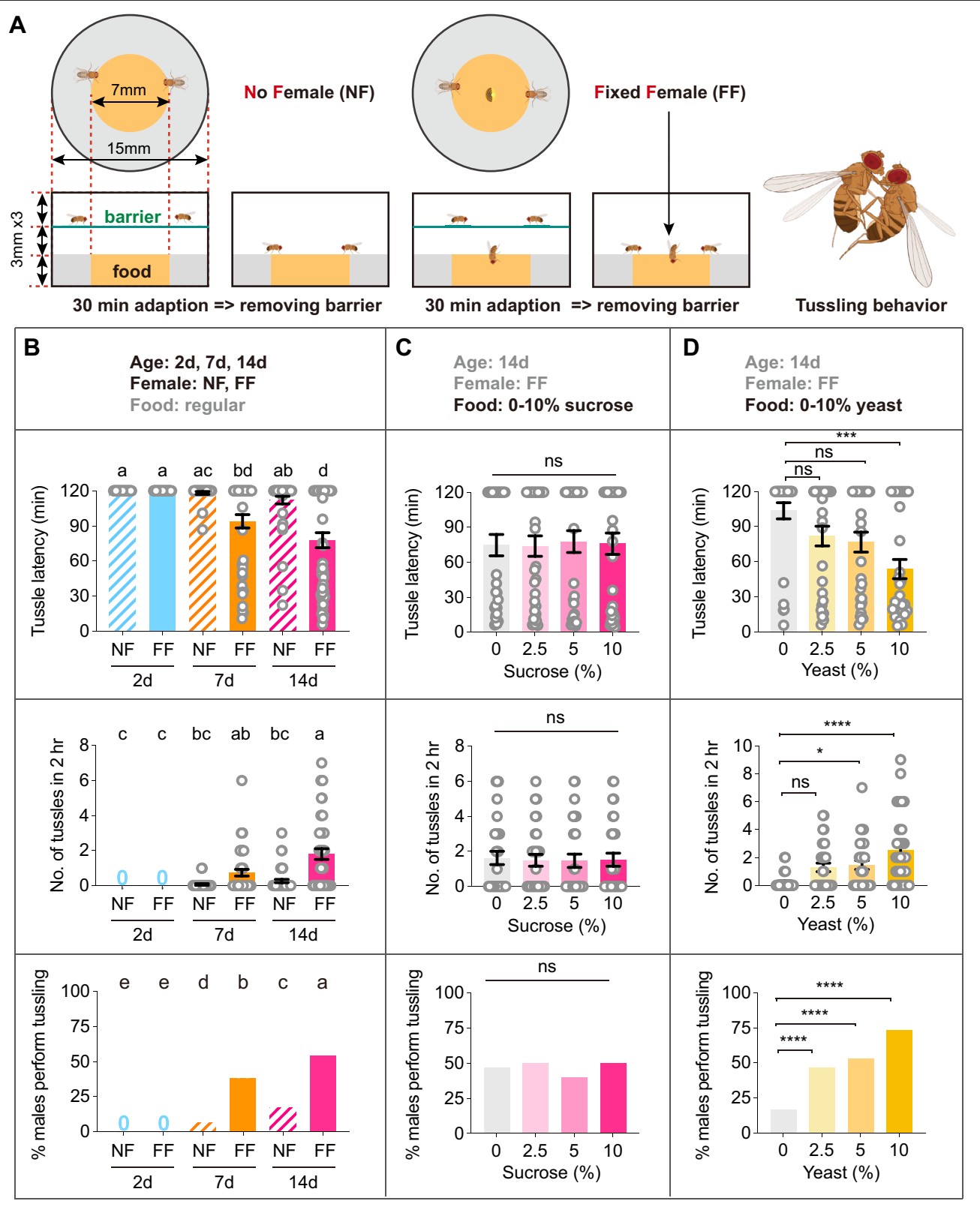

**Figure 1.** A paradigm for tussling behavior in male *Drosophila*. (**A**) Diagrams of two fighting paradigms for *Drosophila* tussling behavior. The left diagram shows a traditional fighting paradigm with no female (NF) in the food area and a modified fighting paradigm with a fixed female (FF) in the center of the food area. Before the experiment began, two male flies were placed on the upper layer, separated from the food by a film. After a 30 minute adaptation, the film was removed to start recording. The right diagram shows a pair of tussling male *Drosophila*. All tester males in this

*Figure 1 continued on next page*

*Figure 1 continued*
figure were group housed. (**B**) The influence of age and female factors on male tussling behavior. From top to bottom, the tussle latency, the number of tussles, and the proportion of tussles occurring are shown respectively. n=30, 30, 30, 46, 46, 46 from left to right. Bars sharing the same letter are not significantly different according to the Kruskal-Wallis test (tussle latency, number of tussles) and Chi-square test (proportion of tussles occurring). (**C**) Sucrose concentration of food has no effect on male tussling behavior. n=30 for each group, n.s., not significant, Kruskal-Wallis test (tussle latency, number of tussles), and Chi-square test (proportion of tussles occurring). (**D**) The influence of yeast concentration on male tussling behavior. n=30 for each group, n.s., not significant, *p<0.05, ***p<0.001, ****p<0.0001, Kruskal-Wallis test (tussle latency, number of tussles), and Chi-square test (proportion of tussles occurring). Error bars correspond to SEM.

The online version of this article includes the following video, source data, and figure supplement(s) for figure 1:

**Source data 1.** Source Data for *Figure 1* on tussling behavior in wild-type males.

**Figure supplement 1.** Aging suppresses lunging behavior in male *Drosophila*.

**Figure supplement 1—source data 1.** Source Data for *Figure 1—figure supplement 1* on lunging behavior in males with different ages or housing conditions.

**Figure 1—video 1.** Tussling behavior between two 14-day-old group-housed (G14) wild-type males.

https://elifesciences.org/articles/104212/figures#fig1video1

behaviors (*Lim et al., 2014*). Therefore, we next assayed tussling behavior in 14-day-old males in the presence of an embedded virgin female with different concentrations of sucrose or yeast in the food during the behavioral test. We found that these males displayed similar levels of tussling behavior with different concentrations of sucrose in the food (*Figure 1C*). On the contrary, male tussling intensity increased significantly with higher yeast concentrations such that more than 70% of males displayed this high-intensity fighting behavior (*Figure 1D*, see *Figure 1—video 1*). Taken together, these results provide an efficient behavioral paradigm for the high-intensity tussling behavior and indicate that older males display more tussling but less lunging than younger ones.

## Social enrichment inhibits low-intensity lunging but enhances high-intensity tussling

Previous studies have shown that social enrichment inhibits aggressive behaviors, specifically the low-intensity lunging, in male flies (*Yadav et al., 2024*). Therefore, we wondered whether social enrichment might also inhibit the high-intensity tussling behavior in males. To answer this question, we compared both fighting behaviors in SH and GH 14-day-old males (*Figure 2A*). Lunging and tussling events from each group of representative samples were exhibited in raster plots for comparison (*Figure 2B*). We scored lunging behavior mainly in the first 10 min of the test, due to its relatively higher occurrence, and tussling behavior during the whole 2 hr observational period since males take much longer time to initiate tussling (*Figure 1D*). Interestingly, we found that while SH males performed more frequent lunging behavior in 10 min than GH males did, which is consistent with previous findings, they rarely showed tussling behavior. In contrast, GH males displayed a large number of tussling events within 2 hr (*Figure 2B*). Statistical analysis of tussling and lunging behaviors showed that GH males exhibited reduced tussling latency and enhanced tussling frequency, in addition to increased lunging latency and decreased lunging frequency, compared to SH males (*Figure 2C–F*). We also quantified the duration of each tussling and lunging event and found that the average duration of individual lunging events was less than 0.2 s, while a tussling event lasted from seconds to minutes (*Figure 2G*). This further suggests that tussling is a more intense form of aggressive behavior compared to lunging. These results indicate that social enrichment inhibits low-intensity lunging but promotes high-intensity tussling in male flies.

## *Or47b* ORNs mediate social experience-induced tussling behavior

To further investigate the sensory basis underlying the opposite regulation of lunging and tussling behaviors by social experience, we tried to examine the role of cVA-sensitive neurons, as well as other *fru*[M]-expressing ORNs, in mediating intermale aggression. Two cVA sensory neurons, *Or67d* and *Or65a* ORNs, play crucial roles in enhancing the lunging-based aggression in SH males, in which *Or67d* ORNs mediate acute cVA stimulation and promote male aggression (*Wang and Anderson, 2010*), while *Or65a* ORNs mediate chronic cVA stimulation and inhibit male aggression in GH males (*Liu et al., 2011*). In addition, the *fru*[M]-expressing *Or47b* ORNs are sensitive to a class of pheromones

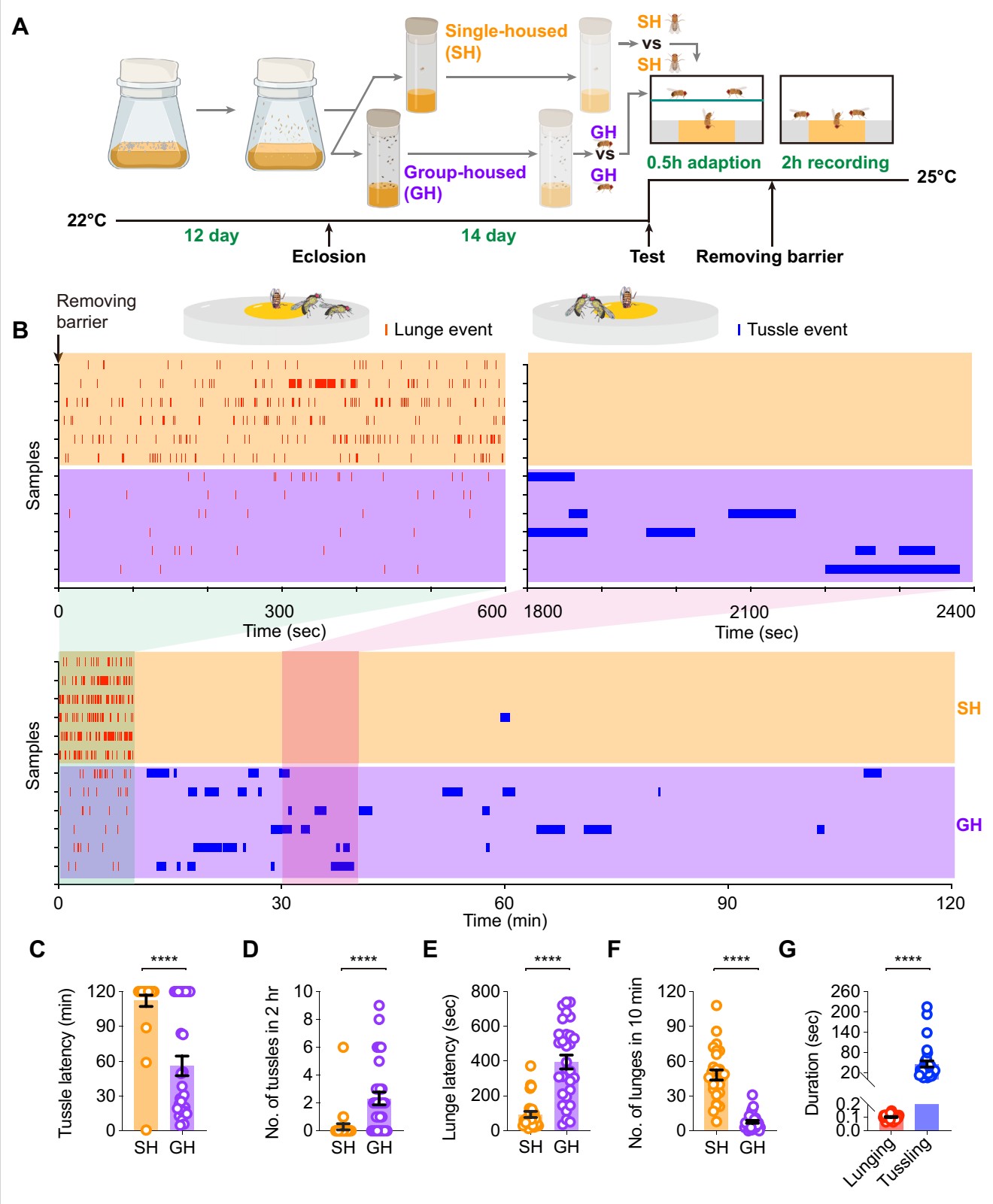

**Figure 2.** Social enrichment inhibits lunging behavior but promotes tussling behavior in males. (**A**) Schematic diagram of *Drosophila* rearing conditions and experimental procedure. (**B**) Representative raster plots illustrating lunge events in the first 10 min and tussle events over 2 hr in single-housed (SH) and group-housed (GH) males. (**C, D**) Latency (**C**) and number (**D**) of tussling behavior in SH and GH males, n=27, 30 for SH and GH males, respectively, ****p<0.0001, Mann-Whitney test for both. (**E, F**) Latency (**E**) and number (**F**) of lunging behavior in SH and GH males, n=27, 30 for SH and GH males,

*Figure 2 continued on next page*

*Figure 2 continued*

respectively, ****p<0.0001, Mann-Whitney test for both. Error bars correspond to SEM. (**G**) Average duration of lunging events in the first 10 min and tussling events within 2 hr in GH males, n=30 events from 10 samples for each group, ****p<0.0001, Mann-Whitney test. Error bars correspond to SEM.

The online version of this article includes the following source data for figure 2:

**Source data 1.** Source Data for *Figure 2* on tussling and lunging behaviors in single-housed and group-housed males.

common to both sexes, such as palmitoleic acid (PA; *Lin et al., 2016*), and their sensitivity is enhanced under group rearing conditions (*Sethi et al., 2019*). *Ir84a* ORNs, which are also *fru*[M]-positive, are involved in the perception of mixed food odors and promote male courtship behavior (*Grosjean et al., 2011*). We silenced *Or67d*, *Or65a*, *Or47b*, or *Ir84a* ORNs by expressing the inwardly rectifying potassium channel Kir2.1 (*Baines et al., 2001*) and assayed lunging and tussling behaviors in 14-day-old GH males. We found that silencing *Or47b* ORNs, but not *Or67d*, *Or65a*, or *Ir84a* ORNs, significantly decreased male tussling, compared to the control (*Figure 3A and B*, see *Supplementary file 1* for detailed genotypes). In contrast, silencing *Or67d* ORNs, but not other ORNs, significantly reduced male lunging (*Figure 3C and D*). These loss-of-function results suggest that *Or47b* ORNs mediate male tussling behavior, whereas *Or67d* ORNs regulate male lunging behavior.

To elucidate the potential role of the Or47b receptor in mediating tussling behavior, we knocked down its expression in *Or47b* ORNs using RNA interference (RNAi). Our results showed a significant decrease in male tussling behavior when the Or47b receptor expression was reduced (*Figure 3E and F*), indicating that the functionality of *Or47b* ORNs in tussling is dependent on the Or47b receptor. As previous studies suggested that the male-specific *fru*[M] sustains *Or47b* expression in an activity-dependent manner (*Sethi et al., 2019*), we next expressed microRNAs targeting *fru*[M] (*UAS-fruMi*; *Meissner et al., 2016*) in *Or47b* ORNs and assayed male tussling. The efficiency of *Or47b* and *fru*[M] RNAi knockdown was both validated by quantitative PCR (*Figure 3—figure supplement 1*). We observed a slight increase in tussling latency in *fru*[M] knocked-down males, although not significantly different from the control (*Figure 3E and F*), suggesting that Or47b, rather than Fru[M], plays a more crucial role in mediating male tussling.

To further confirm that the effect of social enrichment on tussling depends on the activity of *Or47b* ORNs, we artificially activated *Or47b* ORNs by expressing the bacterially derived sodium channel (NaChBac; *Ren et al., 2001*) in 14-day-old SH males. We found that these SH males with enhanced activity of *Or47b* ORNs showed a significant increase in tussling behavior compared to control SH males. Interestingly, the level of tussling in these SH males with activated *Or47b* ORNs was indistinguishable from that in GH males (*Figure 3G and H*). These gain-of-function results indicate that social enrichment acts through *Or47b* ORNs to promote male tussling. Using the trans-Tango technique that labels downstream neurons through ligand-receptor mediated signaling (*Talay et al., 2017*), we found substantially more downstream signals of *Or47b* ORNs in GH males than those in SH males (*Figure 3—figure supplement 2*), suggesting the involvement of *Or47b* ORNs under social enrichment.

## Distinct central brain neurons for lunging and tussling

We next set out to identify the central neurons involved in regulating the high-intensity tussling. We found that silencing the mushroom body neurons by expressing an inwardly rectifying potassium channel Kir2.1 (*Baines et al., 2001*) driven by *R19B03-GAL4* did not affect male tussling, suggesting that the experience-dependent tussling behavior is not dependent on the classical learning and memory center. Surprisingly, inactivation of the previously identified aggression-promoting P1[a] and TK neurons did not affect tussling behavior (*Figure 4A and B*). These results imply that the central circuit for tussling behavior may be distinct from the previously reported aggression circuits. As *dsx*-expressing pC1 neurons, also termed P1 if co-expressing *fru*[M], are crucial for sexual and aggressive behaviors in both males and females, we also silenced two subsets of pC1 neurons (pC1[SS1] and pC1[SS2]), which were sexually dimorphic, and their functions were previously characterized in females but not in males (*Wang et al., 2021*). We found that inactivation of pC1[SS2] but not pC1[SS1] neurons significantly reduced the tussling behavior in male flies (*Figure 4A and B*). The cell bodies of pC1[SS2] and P1[a] neurons are both located in the center of posterior brain and project to overlapping but distinct parts of the lateral protocerebral complex (*Figure 4C–E*). These results suggest that pC1[SS2] neurons may be key central neurons regulating tussling behavior.

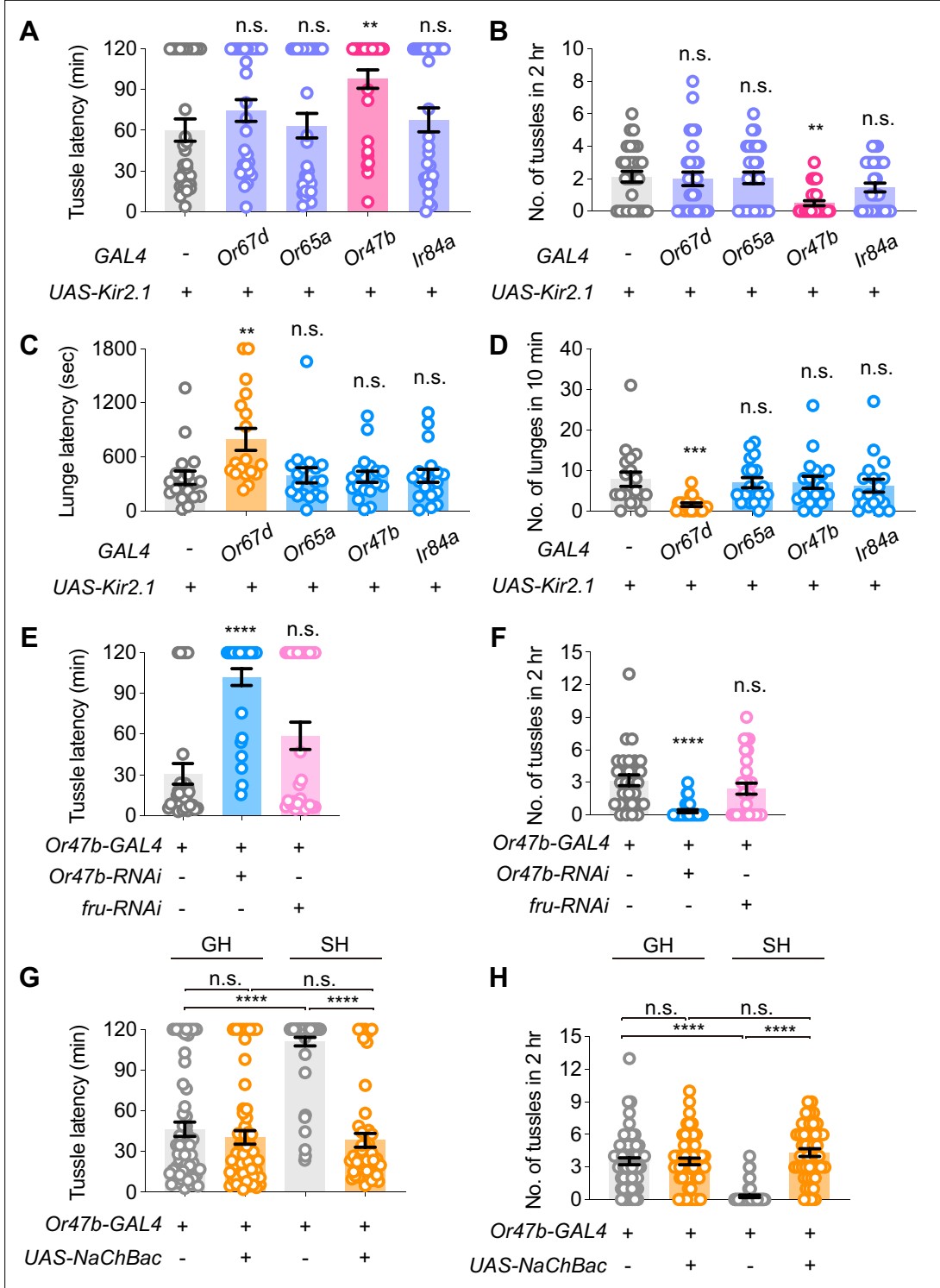

**Figure 3.** Or47b neurons are required for male tussling but not lunging. (**A, B**) The effect of inactivating different sensory neurons on latency (**A**) and number (**B**) of tussling behavior in 14-day-old group-housed males (G14). n=30 for each group, n.s., not significant, **p<0.01, Kruskal-Wallis test. (**C, D**) The effect of inactivating different sensory neurons on the latency (**C**) and number (**D**) of lunging behavior in G14 males. n=18 for each group, n.s., not significant, **p<0.01, ***p<0.001, Kruskal-Wallis test. (**E, F**) The effect of knocking down *Or47b* or *fru$^M$* expression in *Or47b* ORNs on the latency (**E**) and number (**F**) of tussling behavior in G14 males, respectively. n=30 for each group, n.s., not significant, ****p<0.0001, Kruskal-Wallis test. (**G, H**) The effect of activating *Or47b* ORNs on latency (**G**) and number (**H**) of tussling behavior under conditions of GH or SH for 14 days, n=60, 60, 60, 49 respectively, n.s., not significant, ****p<0.0001, Kruskal-Wallis test. Error bars correspond to SEM.

*Figure 3 continued on next page*

*Figure 3 continued*

The online version of this article includes the following source data and figure supplement(s) for figure 3:

**Source data 1.** Source Data for *Figure 3* on tussling and lunging behaviors in Or47b-related transgenic males.

**Figure supplement 1.** Validation of the *Or47b* and *fru^M* RNAi efficiency.

**Figure supplement 1—source data 1.** Source Data for *Figure 3—figure supplement 1* on efficiency of RNAi lines.

**Figure supplement 2.** Group housing increases downstream signals of *Or47b* ORNs.

**Figure supplement 2—source data 1.** Source Data for *Figure 3—figure supplement 2* on trans-Tango signals in single-housed and group-housed males.

To further compare the function of P1$^a$ and pC1$^{SS2}$ neurons in regulating male behaviors, we performed optogenetic activation experiments on these neurons expressing *CsChrimson*. Compared to control male pairs that did not show any significant change upon red light stimulation (*Figure 4F–H*), optogenetic activation of P1$^a$ neurons induced acute courtship behavior as indicated by unilateral wing extension (UWE), and upon red light removal, there was a significant increase in lunging but not tussling (*Figure 4I–K*), consistent with previous findings. In contrast, optogenetic activation of pC1$^{SS2}$ neurons induced strong and acute tussling behavior, with more than 50% of the time during red light in tussling, and the tussling behavior ceased immediately upon removal of red light (*Figure 4L*, see *Figure 4—video 1*). Activation of pC1$^{SS2}$ neurons did not significantly affect either male lunging or courtship (*Figure 4M and N*). These results indicate that pC1$^{SS2}$ neurons acutely promote the high-intensity tussling behavior.

Previous studies have revealed a crucial role of P1$^a$ neurons in promoting a persistent internal state that enhances male aggression (*Hoopfer et al., 2015*). To test whether P1$^a$ neurons can promote both lunging and tussling behaviors, we further performed thermal activation experiments. We found that activation of P1$^a$ neurons via dTrpA1 at 30°C induced male tussling, just like activation of pC1$^{SS2}$ neurons did (*Figure 4—figure supplement 1*). As thermal activation but not optogenetic activation of P1$^a$ neurons induced tussling, we reasoned that vision might be a key factor for this discrepancy, since all optogenetic experiments were performed in dark or red light. Thus, we re-performed these thermal activation experiments in dark and found that activation of pC1$^{SS2}$ but not P1$^a$ neurons induced tussling, although the level of tussling by pC1$^{SS2}$-activated males was lower compared to that in light, suggesting a crucial role of vision in male tussling. These optogenetic and thermogenetic activation experiments suggest that P1$^a$ neurons can promote both lunging and tussling behaviors, but their function on tussling requires visual stimulation, while pC1$^{SS2}$ neurons play a more direct role in acutely promoting male tussling.

We next used *retro*-Tango (*Sorkaç et al., 2023*) and *trans*-Tango (*Talay et al., 2017*) techniques to explore the potential upstream and downstream circuits of P1$^a$ and pC1$^{SS2}$ neurons and observed substantial differences in both conditions (*Figure 4—figure supplement 2*), suggesting largely parallel inputs and outputs for P1$^a$ and pC1$^{SS2}$ neurons.

## Dsx$^M$ regulates the development and function of pC1$^{SS2}$ neurons

To further compare the differences between P1$^a$ neurons and pC1$^{SS2}$ neurons, we investigated whether P1$^a$ and pC1$^{SS2}$ neurons express *fru* and/or *dsx* using Fru$^M$ and Dsx$^M$ antibodies (*Chen et al., 2021*; *Peng et al., 2022*). We found that most P1$^a$ neurons were *fru*-positive and *dsx*-positive, while pC1$^{SS2}$ neurons were *dsx*-positive but *fru*-negative (*Figure 5A–D*). These results suggest that *dsx* instead of *fru* may play crucial roles in the development and function of pC1$^{SS2}$ neurons.

To investigate the function of *dsx* in pC1$^{SS2}$ neurons, we knocked down *dsx* via RNAi and compared the morphological and functional changes of these neurons. We found that *dsx$^M$* knockdown in pC1$^{SS2}$ neurons slightly increased the number of cells from two to three pairs to three to four pairs (*Figure 5E–G*), and significantly lengthened the axonal projections ventrally (*Figure 5H*). We reasoned that these developmental abnormalities may affect the tussling-promoting functions of pC1$^{SS2}$ neurons and tried to activate these neurons with or without knocking down *dsx*. We found that males with wild-type *dsx* function displayed intensive tussling behavior for approximately 50% and 70% of the total activation time, respectively, during the first and second activation periods, with the response returning to baseline levels after the removal of red light (*Figure 5I and K*). However, males with *dsx* knocked down in pC1$^{SS2}$ neurons showed no tussling behavior during the first activation session and

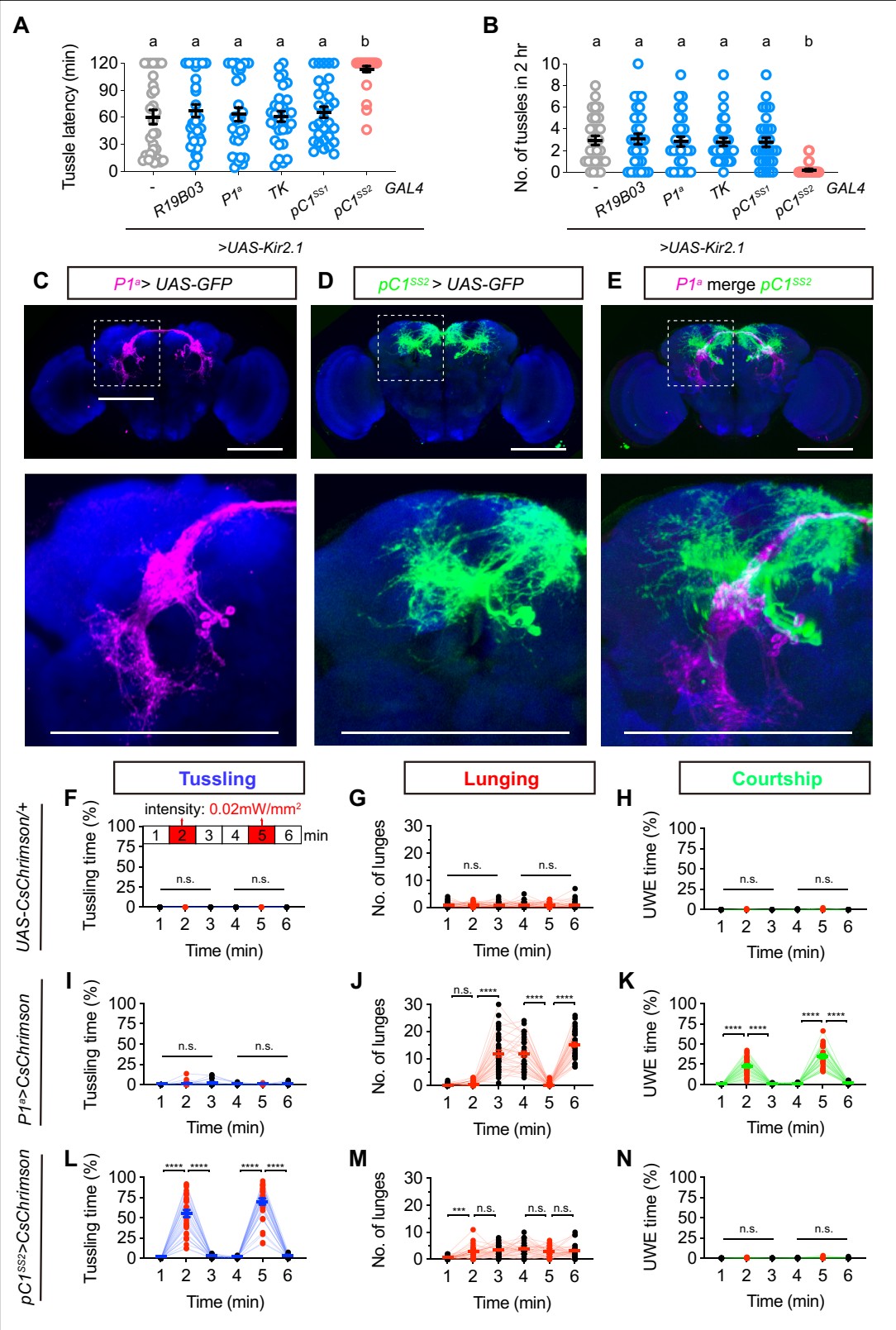

**Figure 4.** A subset of pC1 neurons specifically promotes male tussling behavior. (**A, B**) The effect of inactivating candidate interneurons on latency (**A**) and number (**B**) of male tussling behavior, n=30 for each group, bars sharing the same letter are not significantly different according to Kruskal-Wallis test. (**C–E**) The expression pattern of *P1ᵃ-spGAL4* (**C**) and *pC1ˢˢ²-spGAL4* (**D**) in male brain, and (**E**) the registration of P1ᵃ (magenta) and pC1ˢˢ² (green) neurons in a standard brain. The GFP expression indicates P1ᵃ (magenta) and pC1ˢˢ² (green) neurons in the male brain counterstained with nc82

*Figure 4 continued on next page*

*Figure 4 continued*

(blue). White dashed boxes in (**C–E**) were zoomed in and shown below each panel. Scale bars, 100 µm. (**F–H**) Statistics on tussling (**F**), lunging (**G**), and unilateral wing extension (UWE, **H**) by *UAS-CsChrimson/+* males. Red light activation with a light intensity of 0.02 mW/mm$^2$ for 1 min was applied at the 2nd and 5th minutes of this and subsequent experiments. n=32 for each group, n.s., not significant, Friedman test. (**I–K**) Statistics on tussling (**I**), lunging (**J**), and UWE (**K**) in *P1$^a$>UAS-CsChrimson* males. n=32 for each group, n.s., not significant, ****p<0.0001, Friedman test. (**L–N**) Statistics on tussling (**L**), lunging (**M**), and UWE (**N**) in *pC1$^{SS2}$>UAS-CsChrimson* males. n=32 for each group, n.s., not significant, ***p<0.001, ****p<0.0001, Friedman test.

The online version of this article includes the following video, source data, and figure supplement(s) for figure 4:

**Source data 1.** Source Data for *Figure 4* on tussling, lunging and courtship behaviors in males with manipulated pC1SS2 or P1a neurons.

**Figure supplement 1.** Light-dependent initiation of male tussling behavior.

**Figure supplement 1—source data 1.** Source Data for *Figure 4—figure supplement 1* on tussling behavior in males with dTrpA1-mediated activation of pC1SS2 or P1a neurons.

**Figure supplement 2.** Immunostaining images of potential upstream signals (*retro*-Tango) and downstream signals (*trans*-Tango) of P1$^a$ and pC1$^{SS2}$ neurons.

**Figure 4—video 1.** Optogenetic activation of pC1$^{SS2}$ neurons acutely induces tussling in 5- to 7-day-old group-housed males (*pC1$^{SS2}$>UAS-CsChrimson*).

https://elifesciences.org/articles/104212/figures#fig4video1

few tussling behaviors during the second activation session (*Figure 5J and K*). Together, these results reveal the crucial role of the *dsx* gene in the development and tussling-promoting function of pC1$^{SS2}$ neurons.

## Territory and mating advantage in tussling-favored experienced males

As our results showed that GH males and SH males were more likely to perform the low-frequency, high-intensity tussling and the high-frequency, low-intensity lunging, respectively, we wondered which fighting strategy might be more practical in the competition for territory and mating resources.

To at least partially address this question, we first designed an experimental paradigm for territory competition (*Figure 6A*). A circular territory, which is provided with food and a fixed virgin female and separated by a wall with a small door open to the outer area, was competed by two tester males, one of which was marked on their wings. We manually analyzed the first 10 winning events, each defined as one male successfully displacing the other from the inner resource area. The winning index was calculated as the proportional difference in the number of wins between the marked male and unmarked male. In control groups such as two GH males of the same age, we observed some 'winner takes all' phenomenon (one male won nearly all of the 10 encounters), but overall, the winning index was close to zero due to the randomness of marking (see methods). We further found that while GH 7-day-old males (G7) won slightly, but not significantly, more than SH 7-day-old males (S7) (*Figure 6B*), older G14 males had a significant advantage over the S14 males (*Figure 6C*, *Figure 6—figure supplement 1*, *Supplementary file 2*, and *Figure 6—video 1*). Additionally, we observed that when G14 males were paired with S14 males, the proportion of males performing tussling behavior was not significantly different from that between two G14 males. However, the duration of each tussling event in G14 and S14 male pairs was significantly shorter than that between two G14 males (*Figure 6—figure supplement 2*).

Next, we conducted further investigations into the role of social experience and the two types of fighting behaviors in mating competition (*Figure 6D*). Two males and one virgin female were loaded into the behavioral chamber at the same time and recorded for courtship behaviors. A winner was defined as the male that first copulated with the female. We used the copulation advance index, which was calculated as the proportional difference between marked males and unmarked males in winning copulation (see Materials and methods). We found that G7 males had no significant advantage over S7 males (*Figure 6E*); however, G14 males displayed a significantly increased mating advantage over S14 males (*Figure 6F*, see *Figure 6—video 2*). Together, these results suggest that the tussling-favored GH males are more competitive in territory control and mating compared to the lunging-favored SH males, supporting a correlation between tussling and reproductive competition.

The above results showed that GH males were more competitive compared to SH males, especially in older males, suggesting that social experience may play a potential role in altering age-related mating competition. To test this hypothesis, we first tested mating competition between males of

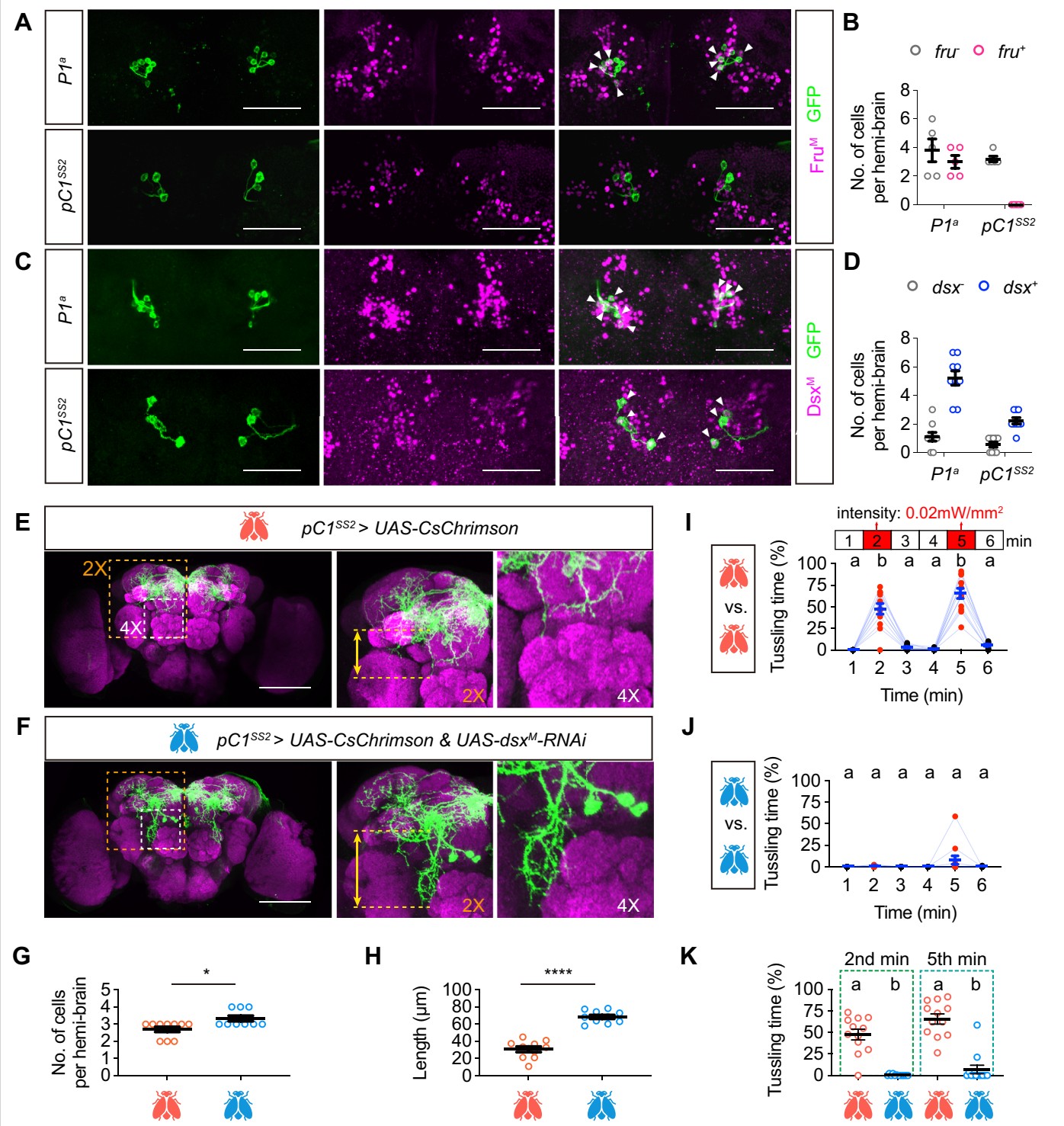

**Figure 5.** Dsx^M regulates the development and tussling-promoting function of pC1^SS2 neurons. (**A, B**) The co-staining of P1^a or pC1^SS2 neurons with the Fru^M antibody (**A**), and the quantification of *fru*-positive neurons in P1^a and pC1^SS2 populations (**B**). n=5 for each group. Scale bars, 50 μm. (**C, D**) The co-staining of P1^a or pC1^SS2 neurons with the Dsx^M antibody, and the quantification of *dsx*-positive neurons in P1^a and pC1^SS2 populations (**D**), n=9 for each group. Scale bars, 50 μm. (**E–K**) Knocking down *dsx* in pC1^SS2 neurons impacts their development and tussling-promoting function. (**E, F**) Immunostaining of pC1^SS2 neurons in control (**E**) and *dsx* knock-down (**F**) males. Magenta represents nc82 signals, and green represents GFP signals. The images within the yellow dashed box and white dashed box are magnified ×2 and ×4, respectively, and displayed on the right. Scale bars, 100 μm. (**G**) Quantification of the number of pC1^SS2 neurons in control and *dsx* knock-down males. n=10, 9 for control and experimental group respectively, *p<0.05, Mann-Whitney test. Error bars correspond to SEM. (**H**) Quantification of the neuronal projection length shown in the ×2 magnified images in

*Figure 5 continued on next page*

*Figure 5 continued*

(**E**) and (**F**). n=10 for each group, ****p<0.0001, Mann-Whitney test. (**I–K**) Tussling time by *pC1^{SS2}SS2UAS-CsChrimson* (**I**) and *pC1^{SS2}SS2UAS-CsChrimson & UAS-dsx^{M}-RNAi* (**J**) males, which were quantified in (**K**). Red light activation with a light intensity of 0.02 mW/mm² for 1 min was applied at the 2nd and 5th minutes. n=12, bars sharing the same letter are not significantly different. Friedman test for (**I**) and (**J**), Kruskal-Wallis test for (**K**).

The online version of this article includes the following source data for figure 5:

**Source data 1.** Source Data for *Figure 5* on how DsxM affects the development and function of pC1SS2 neurons.

different ages under the same rearing conditions. The results showed that younger males, whether reared in group or isolation, had a greater mating advantage compared to older males (G7 >G14, S7 >S14, G14 >G21, S14 >S21; *Figure 6G*, *Supplementary file 2*). These results indicate that there is generally a disadvantage in mating competition associated with aging. We next investigated whether the mating advantage induced by social experience could compensate for the mating disadvantage caused by aging. We tested pairs of males with different ages and rearing conditions (G7 vs. S14, S7 vs. G14, G14 vs. S21, S14 vs. G21) for mating competition. We found that while 14-day-old males were less competitive than 7-day-old males regardless of the rearing condition, indicative of the aging-related disadvantage, 21-day-old GH males were significantly more competitive than 14-day-old SH males (G21 >S14; *Figure 6H*).

To further investigate whether the territorial control and mating competition advantage in GH males are related to their high tussling behavior, we paired males with low tussling behavior (*Or47b-GAL4/ UAS-Kir2.1* and *pC1^{SS2}/UAS-Kir2.1*) with control males for territorial control and mating competition tests. We found that males with inactivated Or47b or pC1^{SS2} neurons were incompetent for both territorial control and mating competition compared to control males (*Figure 6—figure supplement 3*), further supporting a correlation between the occurrence of tussling behavior and the acquisition of territorial and mating competition advantages.

Taken together, these results indicate that long-term social enrichment in older males increases their territorial and mating competition, which can overcome their mating disadvantage associated with aging (*Figure 7*).

## Discussion

Regulation of physiology and behavior by social experience is an important feature across animal species. Here, we found that social experience shapes the fighting strategies of male flies, in which GH males tend to reduce the low-intensity lunging and increase the high-intensity tussling. We further showed that the two forms of fighting are mediated by distinct sensory and central neurons. The complex implications of these findings are discussed below.

Previous studies have found that social isolation generally enhances aggression but decreases mating competition in animal models (*Palavicino-Maggio and Sengupta, 2022*; *Sethi et al., 2019*; *Yadav et al., 2024*), presenting a paradox between social experience, aggression, and reproductive success. Our results provide an explanation for this paradox by dissecting aggressive behaviors into the well-studied, frequently observed lunging behavior and the less frequent but high-intensity tussling behavior. We improved the behavioral paradigm for studying tussling behavior and revealed that social experience oppositely regulates lunging and tussling. It is important to note that the frequency of lunging decreases while the frequency of tussling increases from 7-day-old to 14-day-old males, suggesting a shift in fighting strategy with age. Our results further suggest that social experience may enhance male mating competition by shifting their fighting strategy to the low-frequency, high-intensity tussling, which may be more effective in combat, although direct evidence is still lacking. These results are consistent with a previous study showing a correlation between the establishment of male dominance and tussling behavior, but not lunging (*Simon and Heberlein, 2020*). Interestingly, we found that the mating advantage gained through social enrichment can even offset the mating disadvantage associated with aging. Note that a previous study found a mating advantage in 7-day-old males compared to 2-day-old males (*Lin et al., 2016*), which may be due to different levels of sexual maturity, but not necessarily reflect the aging effect.

Our results revealed distinct neural circuits underlying the two forms of fighting, spanning from primary sensory neurons to central regulatory neurons. Specifically, silencing the cVA-sensing Or67d neurons reduced male lunging but not tussling, whereas inactivating Or47b neurons impaired male

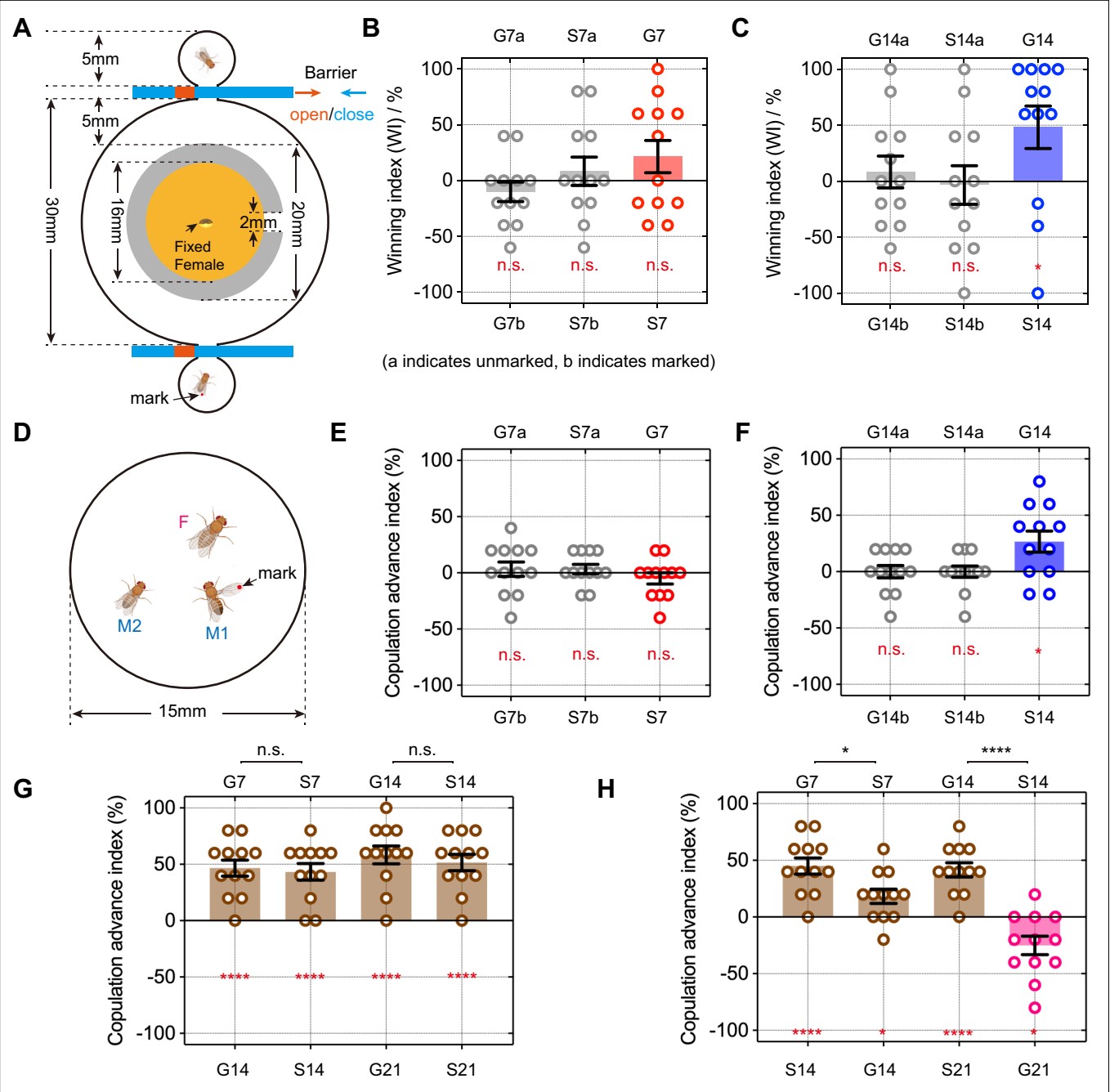

**Figure 6.** Social enrichment enhances male territorial control and mating competition. (**A**) Schematic diagram of the paradigm for testing territorial control in males. (**B**) The territorial control test of 7-day-old GH (G7) and SH (S7) males. G7a and G7b stand for marked G7 male and unmarked G7 male, respectively. n=12 for each group, n.s., not significant, one sample *t*-test. (**C**) The territorial control test of 14-day-old GH (G14) and SH (S14) males. n=12 for each group, n.s., not significant, *p<0.05, one sample *t* test. (**D**) Schematic diagram of the paradigm for testing male mating advantage. (**E**) The mating competition test between G7 and S7 males. n=12 for each group, n.s., not significant, one sample *t* test. (**F**) The mating competition test between G14 and S14 males. n=12 for each group, n.s., not significant, *p<0.05, one sample *t* test. (**G** and **H**) The mating competition test between young males and old males, n=12 for each group, n.s., not significant, *p<0.05, ****p<0.0001, one sample *t* test for comparison within group (red labeling), Mann-Whitney test for comparison between groups (black labeling). Error bars correspond to SEM.

The online version of this article includes the following video, source data, and figure supplement(s) for figure 6:

**Source data 1.** Source Data for *Figure 6* on winning index and copulation advance index in wild-type males.

**Figure supplement 1.** Winning events of representative samples for territorial competition.

**Figure supplement 2.** The occurrence and duration of tussling behavior in male flies under different pairing conditions.

*Figure 6 continued on next page*

*Figure 6 continued*

**Figure supplement 2—source data 1.** Source Data for *Figure 6—figure supplement 2* on tussling behavior between single-housed and group-housed males.

**Figure supplement 3.** Silencing Or47b or pC1^SS2 neurons compromises male territorial control and mating competition.

**Figure supplement 3—source data 1.** Source Data for *Figure 6—figure supplement 3* on winning index and copulation advance index in males with pC1SS2 or Or47b neurons being silenced.

**Figure 6—video 1.** In a territory competition paradigm, a 14-day-old group-housed (G14, marked) wild-type male consistently outcompetes a 14-day-old single-housed (S14, unmarked) wild-type male.

https://elifesciences.org/articles/104212/figures#fig6video1

**Figure 6—video 2.** In a mating competition paradigm, a 14-day-old group-housed (G14, unmarked) wild-type male outperformed a 14-day-old single-housed (S14, marked) wild-type male and successfully copulated with the female.

https://elifesciences.org/articles/104212/figures#fig6video2

tussling but not lunging. Notably, Or47b neurons respond to pheromones common to both sexes, and the sensitivity of these sensory neurons is enhanced by social enrichment (*Lin et al., 2016*; *Peng et al., 2021*; *Sethi et al., 2019*). Unlike the Or67d-dependent lunging, which is primarily male-induced and socially inhibited, the Or47b-dependent tussling is promoted by social enrichment with either male or female flies. Recently, it has been found that female flies increase aggression towards mating pairs or mated females, depending on Or47b and Or67d, respectively (*Gaspar et al., 2022*; *Wan et al.,*

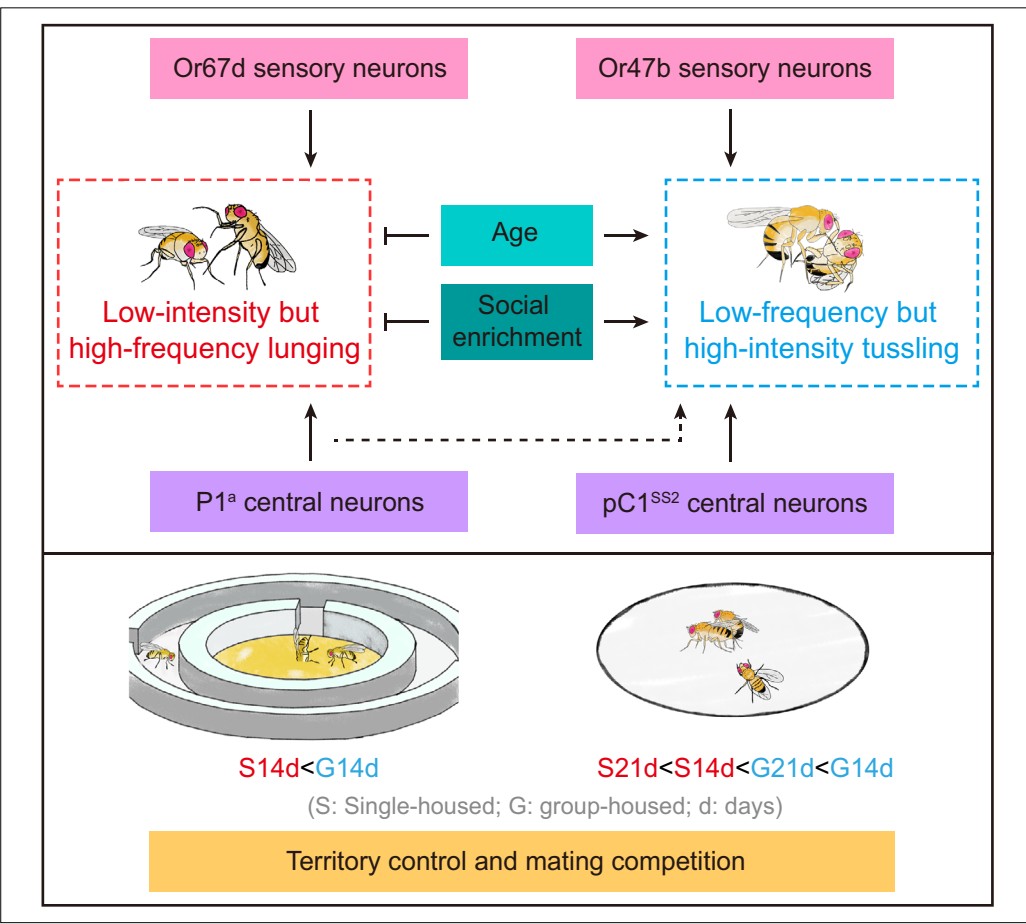

**Figure 7.** Summary model of fighting strategies and reproductive success in male *Drosophila*. Age and social enrichment inhibit the low-intensity, high-frequency lunging while promoting the low-frequency, high-intensity tussling. This shift in fighting strategies among experienced, aged males is accompanied by better territorial control and mating competition, even offsetting aging-related mating disadvantages. Lunging and tussling behaviors are regulated by distinct sensory and central neurons.

*2023*). These results suggest a common function of Or47b and Or67d neurons in both sexes for regulating aggression. However, whether these neurons mediate different fighting patterns in females and how they respond to social experience in females remain unknown. According to the FlyWire connectome database (*Dorkenwald et al., 2024*; *Schlegel et al., 2024*), connecting Or47b and Or67d ORNs to pC1 neurons in females requires at least two intermediate neurons, suggesting a high degree of parallel processing from ORNs to pC1s.

Regarding central regulatory neurons, we identified a novel class of pC1 neurons, pC1SS2, that specifically promote male tussling. In male flies, there are ~60 pairs of *dsx*-expressing pC1 neurons, including the well-characterized P1a neurons that also express *fruM* (*Asahina, 2018*; *Jiang and Pan, 2022*); however, there are only three pairs of pC1SS2 neurons that specifically express *dsx* but not *fruM* and are crucial for the high-intensity tussling in males. The three pairs of pC1SS2 neurons may have distinct functions from P1a neurons in several ways. (1) P1a neurons promote both male courtship and aggression (*Hoopfer et al., 2015*; *Zhao et al., 2024*), whereas pC1SS2 neurons specifically promote aggression but not courtship. A recent study also identified pC1SS2 neurons as aggression-promoting, but their roles in different fighting modes are not depicted (*Hindmarsh Sten et al., 2025*). That tussling-promoting pC1SS2 neurons specifically express *dsx* but not *fruM* is also consistent with a previous finding that *dsx*-positive pC1 neurons, rather than *fruM*-positive pC1 neurons, are crucial for male aggression (*Koganezawa et al., 2016*). Moreover, the anatomical features of pC1SS2 neurons are highly similar to the male-specific aggression-promoting (MAP) neurons identified previously (*Chiu et al., 2021*). (2) P1a neurons promote both lunging and tussling, although their function on tussling requires additional visual stimulation. In contrast, pC1SS2 neurons specifically promote tussling, and the presence of light enhances but is not absolutely necessary for tussling. These results align with previous findings that visual stimuli amplify P1 activation and related behaviors (*Hindmarsh Sten et al., 2021*; *Kohatsu and Yamamoto, 2015*; *Pan et al., 2012*), and visual stimulation is important for male aggression (*Duistermars et al., 2018*; *Ramin et al., 2014*; *Schretter et al., 2020*). (3) Male lunging is induced after P1a activation, whereas male tussling is induced during pC1SS2 activation. These results suggest that P1a neurons may not directly regulate aggression but instead induce a persistent state that promotes aggression in general, while pC1SS2 neurons may play a more direct role in mediating tussling. Future studies are needed to further dissect the seemingly parallel sensory and central neuronal pathways involved in lunging and tussling and to explore how these pathways interact during social experience and aging to optimize reproductive success.

### Limitations of the study

We manually analyzed aggressive behaviors and strived to ensure objectivity in behavioral analysis; however, precisely tracking and quantifying specific occurrences of lunging and tussling events for each fly over extended time periods remained methodologically challenging. Furthermore, tussling events often exhibit prolonged durations, which may also include other forms of aggression such as holding. While our results indicate an association between tussling behavior and reproductive competition, a direct link between them is lacking. Future investigations are needed to establish a causal relationship between fighting strategies and reproductive success.

## Materials and methods

**Key resources table**

| Reagent type (species) or resource | Designation | Source or reference | Identifiers | Additional information |
|---|---|---|---|---|
| Genetic reagent (*D. melanogaster*) | Wild-type (*Canton-S*) | Janelia Research Campus (*Pan and Baker, 2014*) | | |
| Genetic reagent (*D. melanogaster*) | Or67d-GAL4 | Bloomington *Drosophila* Stock Center | BDSC_9998 | |
| Genetic reagent (*D. melanogaster*) | Or65a-GAL4 | Bloomington *Drosophila* Stock Center | BDSC_9993 | |
| Genetic reagent (*D. melanogaster*) | Or47b-GAL4 | Bloomington *Drosophila* Stock Center | BDSC_9984 | |

*Continued on next page*

*Continued*

| Reagent type (species) or resource | Designation | Source or reference | Identifiers | Additional information |
|---|---|---|---|---|
| Genetic reagent (*D. melanogaster*) | *IR84a-GAL4* | Bloomington *Drosophila* Stock Center | BDSC_41734 | |
| Genetic reagent (*D. melanogaster*) | *TK-GAL4* | Bloomington *Drosophila* Stock Center | BDSC_51795 | |
| Genetic reagent (*D. melanogaster*) | *R19B03-GAL4* | Bloomington *Drosophila* Stock Center | BDSC_49830 | |
| Genetic reagent (*D. melanogaster*) | *UAS-dTrpA1* | Bloomington *Drosophila* Stock Center | BDSC_26263 | |
| Genetic reagent (*D. melanogaster*) | *UAS-Kir2.1* | Bloomington *Drosophila* Stock Center | BDSC_6595 | |
| Genetic reagent (*D. melanogaster*) | *UAS-CsChrimson* | Bloomington *Drosophila* Stock Center | BDSC_82181 | |
| Genetic reagent (*D. melanogaster*) | *UAS-myrGFP* | Bloomington *Drosophila* Stock Center | BDSC_32198 | |
| Genetic reagent (*D. melanogaster*) | *retro*-Tango | Bloomington *Drosophila* Stock Center | BDSC_99661 | |
| Genetic reagent (*D. melanogaster*) | *trans*-Tango | Bloomington *Drosophila* Stock Center | BDSC_77124 | |
| Genetic reagent (*D. melanogaster*) | *UAS-NaChBac* | *Ren et al., 2001* | | |
| Genetic reagent (*D. melanogaster*) | *fru$^{GAL4}$* | *Stockinger et al., 2005* | | |
| Genetic reagent (*D. melanogaster*) | *P1$^a$-spGAL4* | *Zhang et al., 2018* | | |
| Genetic reagent (*D. melanogaster*) | *pC1$^{SS1}$-spGAL4* | *Wang et al., 2021* | | |
| Genetic reagent (*D. melanogaster*) | *pC1$^{SS2}$-spGAL4* | *Wang et al., 2021* | | |
| Genetic reagent (*D. melanogaster*) | *UAS-dsx$^M$-RNAi* | *Peng et al., 2022* | | |
| Genetic reagent (*D. melanogaster*) | *UAS-fruMi* | *Meissner et al., 2016* | | |
| Genetic reagent (*D. melanogaster*) | *UAS-Or47b RNAi* | TsingHua Fly Center | THU_2599 | |
| Antibody | Mouse monoclonal anti-HA | BioLegend | Cat# 901501 | 1:200 |
| Antibody | Rabbit polyclonal anti-GFP | Invitrogen | Cat# A11122 | 1:1000 |
| antibody | Mouse monoclonal anti-GFP | Sigma-Aldrich | Cat# G6539 | 1:500 |
| Antibody | Mouse monoclonal anti-Bruchpilot (nc82) | Developmental Studies Hybridoma Bank | RRID:AB_2314866 | 1:50 |
| Antibody | Rabbit polyclonal anti-Dsx$^M$ | *Peng et al., 2022* | | 1:500 |
| Antibody | Mouse monoclonal anti-Fru$^M$ | *Chen et al., 2021* | | 1:2000 |
| Antibody | Donkey anti-mouse IgG Alexa Fluor 488 | Invitrogen | Cat# A21206 | 1:500 |
| Antibody | Donkey anti-mouse IgG Alexa Fluor 555 | Invitrogen | Cat# A31570 | 1:500 |
| Antibody | Donkey anti-rabbit IgG Alexa Fluor 488 | Invitrogen | Cat# A21206 | 1:500 |
| Antibody | Donkey anti-rabbit IgG Alexa Fluor 555 | Invitrogen | Cat# A31572 | 1:500 |

*Continued on next page*

*Continued*

| Reagent type (species) or resource | Designation | Source or reference | Identifiers | Additional information |
|---|---|---|---|---|
| Commercial assay or kit | SuperScript IV | Invitrogen | Cat# 18091050 | |
| Commercial assay or kit | AceQ qPCR SYBR Green Master Mix | Vazyme | Cat# Q121-02 | |
| Chemical compound, drug | Paraformaldehyde (PFA) | Sigma–Aldrich | CAS# 30525-89-4 | |
| Chemical compound, drug | TRIzol reagent | Invitrogen | Cat# 15596026 | |
| Chemical compound, drug | all *trans*-Retinal | Sigma–Aldrich | MFCD00001550 | |
| Software, algorithm | Prism 9 | GraphPad | https://www.graphpad.com | |
| Software, algorithm | LifeSongY 2.1 | *Guo, 2021*; Written by Dr. Chao Guo based on LifesongX | https://sourceforge.net/projects/lifesongy/ | |
| Software, algorithm | Fiji | NIH, USA | https://imagej.net/Fiji | |
| Software, algorithm | ZEISS ZEN | ZEISS | https://www.zeiss.com.cn/corporate/home.html | |

## Fly stocks

Flies were maintained at 22 or 25°C in a 12 hr:12 hr light: dark cycle. *Canton-S* flies were used as the wild-type strain and obtained from Janelia Research Campus (*Pan and Baker, 2014*). *Or67d-GAL4* (BDSC_9998), *Or65a-GAL4* (BDSC_9993), *Or47b-GAL4* (BDSC_9984), *IR84a-GAL4* (BDSC_41734), *TK-GAL4* (BDSC_51795), *R19B03-GAL4* (BDSC_49830), *UAS-dTrpA1* (BDSC_26263), *UAS-Kir2.1* (BDSC_6595), *UAS-CsChrimson* (BDSC_82181), *UAS-myrGFP* (BDSC_32198), *retro*-Tango (BDSC_99661) and *trans-Tango* (BDSC_77124) flies were obtained from Bloomington *Drosophila* Stock Center (BDSC). *UAS-NaChBac* (*Ren et al., 2001*), *fru^GAL4^* (*Stockinger et al., 2005*), *P1^a^-spGAL4* (*Zhang et al., 2018*), *pC1^SS1^-spGAL4* (*Wang et al., 2021*), *pC1^SS2^-spGAL4* (*Wang et al., 2021*), *UAS-dsx^M^-RNAi* (*Peng et al., 2022*) and *UAS-fruMi* (*Meissner et al., 2016*) were used as previously described. *UAS-Or47b RNAi* (THU_2599) was obtained from TsingHua Fly Center. Detailed genotypes of flies were listed in *Supplementary file 1*.

## Male aggression assay

For the aggression assays presented in *Figure 1*, the prepared food was evenly spread into the center of behavioral chambers. There was no female in the traditional paradigm (*Figure 1A*, left; *Wu et al., 2020*; *Zhou et al., 2008*), while a 3- to 5-day-old virgin female was gently fixed in the center of the food with its abdomen outside in the modified paradigm (*Figure 1A*, right). We used a fixed female to restrict its position in the center of food. Then two male testers were placed into the upper chambers separated from the food (with or without a fixed female) by a film barrier. All testers were loaded by cold anesthesia. After a 30-min adaptation, the film was gently removed to allow the two males to fall into the behavioral chamber, and the aggressive behavior was recorded for 2 hr. The aggression assays in *Figure 1—figure supplement 1* were performed by using the paradigm without female, while other data were acquired using the modified paradigm with a fixed female unless otherwise stated. In some experiments, both single-housed and group-housed (30–40 males per vial) males were assayed for comparison.

We manually analyzed lunging and tussling behaviors in this study. Lunging is characterized by a male raising its forelegs and quickly striking the opponent, and each lunge typically lasts less than 0.2 s through detailed slow-motion analysis. Tussling is characterized by both males using their forelegs and bodies to tumble over each other, and this behavior may last from seconds to minutes. Tussling is often mixed with boxing, in which both flies rear up and strike the opponent with forelegs. Since boxing is often transient and difficult to distinguish from tussling, we referred to the mixed boxing and tussling behavior simply as tussling. As we manually analyze tussling for 2 hr for each pair of males, it is possible that we may miss some tussling events, especially those quick ones.

For the quantification of lunge latency and frequency in *Figures 2E, F and 3C, D*, *Figure 1—figure supplement 1A, B*, we manually analyzed the videos and defined the lunge latency as the time point at which the first lunging event occurred between the two male flies after the start of the recording. We then counted the number of lunge events occurring within 10 min after the first lunge. For the

quantification of tussle latency and frequency in *Figures 1B, C, D, 2C, D, 3A, B, E, F, G, H, 4A and B*, we manually analyzed the videos and defined the tussle latency as the time point at which the first tussling event occurred between the two male flies after the start of the recording. We defined the tussle frequency (No. of tussles in 2 hr) as the total number of tussling events that occurred within two hours from the beginning of the video.

The food recipes for the male aggression assay were listed below. A mixture of 2.5% yeast, 2.5% sucrose, and 1% agar dissolved in apple juice (Huiyuan Apple juice concentrate) was used for *Figure 1B*; a mixture of 2.5% yeast, different concentrations of sucrose (0, 2.5%, 5%, or 10%), and 1% agar dissolved in apple juice was used for *Figure 1C*; a mixture of different concentrations of yeast (0, 2.5%, 5%, or 10%), 2.5% sucrose, and 1% agar dissolved in apple juice was used for *Figure 1D*; A mixture of 10% yeast and 1% agar dissolved in apple juice was used for other experiments.

## dTrpA1-mediated activation experiments

For the thermogenetic activation assay shown in *Figure 4—figure supplement 1*, two 5–7 days old males were loaded into the circular chamber (d=15 mm) under cold anesthesia and separated with a transparent film. Before the recording started, the chamber was transferred to the recording platform at a specific temperature (22 °C or 30 °C) for 30 min adaptation. After that, the film was removed, and the behavior was recorded for 1 hr. The 10-min video after film extraction was used for behavioral analysis.

## CsChrimson-mediated optogenetic activation experiments

For optogenetic activation assay shown in *Figures 4 and 5*, males were raised at 25°C on standard fly food, collected after eclosion and housed in the dark for 5–7 days on food containing 0.4 mM all *trans*-Retinal (MFCD00001550, Sigma-Aldrich, St. Louis, MO). The specific light stimulation protocols are shown in the schematic (*Figure 4F*). In brief, two 5–7 days old males were loaded into the circular chamber (d=10 mm) under freeze anesthesia and separated with a transparent film in the dark. Then the chamber was moved on the plane light source (IR channel: 865 nm, red-light channel: 630 nm) for 15 min adaptation. Before video recording, the IR light channel was switched on and adjusted to the appropriate intensity for recording. The film was removed before the recording started, with the red-light channel switched on (constant red light, 0.02 mW/mm²) during the 2nd and 5th minutes of the experiment. The statistics for the tussling, lunging, and unilateral wing extension (UWE) are as follows. For tussling time and UWE time, we used LifesongY 2.1 software (*Guo, 2021*), which is written by Dr. Chao Guo based on LifesongX (*Bernstein et al., 1992*), to calculate the percentage of the total time these behaviors occurred per minute. Specifically, the 'tussling time (%)' refers to the percentage of time during the observation period in which tussling behavior occurred, relative to the total observation time. Similarly, 'UWE time (%)' represents the percentage of time during the observation period in which UWE occurred. For lunges, we counted the number of lunging occurrences per minute.

## Territorial competition experiment

For the territorial competition experiment in *Figure 6A–C*, *Figure 6—figure supplement 3A*, the behavioral setup was produced using a 3D printer (Bambu Lab A1), as illustrated in the schematic, with a transparent cover plate placed on the top. The preparation of food and female was the same as the above aggression paradigm. The female was fixed in the center of the food, and two male testers were placed separately in the two small chambers on both sides. Before initiating the experiment, the two barriers were closed. After a 30-min adaptation, the barriers were opened to allow the males access to the central large chamber and were closed once the males went inside, and the territorial behavior was recorded for 2 hr. The first 10 encounters occurring after 1 hr of recording were used for statistical analysis. The winning event was defined as a male driving the other male out of the food area. To discriminate between tester males, about half of them were marked on one of their wings using a red marker pen 24 hr before the experiment. For experiments using males with different ages and/or housing conditions, we calculated the winning index as (num. of wins by males [upper panel] – num. of wins by males [lower panel])/10 encounters * 100%. For control experiments using males of the same age and housing condition, we calculated the winning index as (num. of wins by unmarked males – num. of wins by marked males)/10 encounters * 100%, which is roughly zero due to the randomness of marking.

## Mating competition experiment

For the mating competition experiment shown in *Figure 6D–H*, *Figure 6—figure supplement 3B*, the wings of males were marked 24 hr prior to the experiment the same as described above. During the experiment, a 3–5 days old virgin female was placed into the chamber under freeze anesthesia and covered with a transparent film. Then the unmarked males were loaded into the chamber in the same manner and also covered with a transparent film. Finally, the marked males were loaded. After 30 min adaptation, two films were removed to allow the three flies to interact, and the behavior was recorded for 1 hr. The chambers without successful copulation during the 1 hr period were excluded from further analysis. For each condition, we used 120 efficient chambers with 10 random chambers collectively as one sample. For experiments using males with different ages and/or housing conditions, the copulation advance index was defined as (number of male winners [upper panel] – number of male winners [lower panel])/10 chambers * 100%. For control experiments using males of the same age and housing condition, the copulation advance index was calculated as (number of winners by unmarked males – number of winners by marked males)/10 chambers * 100%, which is roughly zero due to the randomness of marking. The winner was defined as the male that first copulated with the female.

## Quantitative real-time PCR

Adult fly samples were frozen in liquid nitrogen and decapitated by vortex. The heads were then separated from the bodies using metal sieves. Each sample, consisting of 40 frozen heads, was used for total RNA preparation using TRIzol reagent (15596026, Thermo Fisher Scientific, Waltham, MA). We purified total RNA using a DNA-free Kit (AM1906, Thermo Fisher Scientific, Waltham, MA) and performed reverse transcription using SuperScript IV (18091050, Thermo Fisher Scientific, Waltham, MA) to obtain cDNA used for templates. Quantitative PCR was performed on the Roche LightCycler 96 Real-Time PCR machine using AceQ qPCR SYBR Green Master Mix (Q121-02, Vazyme, Nanjing). Transcript levels were analyzed by the $2^{-\Delta CT}$ method using *actin* as control. Each sample was run in triplicate. Each experiment was repeated three times using independent sets of genetic crosses.

Primers used for RT-qPCR quantification were:

> *actin* forward: 5'-GTCGCGATTTAACCGACTACCTGA-3'
> *actin* reverse: 5'-TCTTGCTT CGAGATCCACATCTGC-3'
> *fru*-P1 forward: 5'-GTGTGCGTACGTTTGAGTGT-3'
> *fru*-P1 reverse: 5'-TAATCCTGTGACGTCGCCAT-3'
> *Or47b* forward: 5'-CAAATCTCAGCCTTCTGCGG-3'
> *Or47b* reverse: 5'-GATACTGGCACAGCAAACTCA-3'

## Tissue dissection, staining, and imaging

Brains were dissected in Schneider's insect medium (S2) and fixed in 4% paraformaldehyde in 0.5% Triton X-100 and 0.5% bovine serum albumin in phosphate-buffered saline (PAT) for 30 min at room temperature. After 4 × 10 min washes, tissues were blocked in 3% normal goat serum (NGS) for 60 min, then incubated in primary antibodies diluted in 3% NGS for 4 h at room temperature and 1 day at 4°C, then washed in PAT, and incubated in secondary antibodies diluted in 3% NGS for 4 h at room temperature and 1 day at 4°C. The tissues were then washed thoroughly in PAT and mounted for imaging. Primary antibodies used in this study include mouse anti-HA (BioLegend, 901501, 1:200), rabbit anti-GFP (Invitrogen, A11122, 1:1000), monoclonal mouse anti-GFP (Sigma-Aldrich, G6539, 1:500), mouse anti-Bruchpilot (Developmental Studies Hybridoma Bank, nc82, 1:50), rabbit anti-Dsx$^M$ (1:500) and mouse anti-Fru$^M$ (1:2000). Secondary antibodies used include donkey anti-mouse IgG conjugated to Alexa 488 (Invitrogen, A21206, 1:500), donkey anti-mouse IgG conjugated to Alexa 555 (Invitrogen, A-31570, 1:500), donkey anti-rabbit IgG conjugated to Alexa 488 (Invitrogen, A-21206, 1:500) and donkey anti-rabbit IgG conjugated to Alexa 555 (Invitrogen, A-31572, 1:500). Samples were imaged on Zeiss 900 confocal microscopes using ZEN and processed with Fiji software.

## Trans-synaptic labeling by *trans*-Tango and *retro*-Tango

The *trans*-Tango or *retro*-Tango ligand with GFP was expressed under the GAL4 driver. Trans-synaptic signals were marked by a membrane-associated form of tdTomato (QUAS-mtdTomato-3xHA). The anti-HA antibody (BioLegend, 901501, 1:200) was applied to characterize the postsynaptic or

presynaptic signal. For *Figure 3—figure supplement 2*, male testers were kept at 18 °C until eclosion and subsequently single- or group-housed at the same temperature for two weeks before dissection and imaging. For *Figure 4—figure supplement 2*, flies were kept at 18 °C for *trans*-Tango and 25 °C for *retro*-Tango experiments.

## Brain image registration

The protocol for standard brain registration is described previously (*Zhou et al., 2014*). Confocal images for brains containing P1ᵃ neurons and pC1^SS2 neurons were registered onto a standard reference brain (*Zhou et al., 2014*), using a Fiji graphical user interface as described previously (*Ostrovsky et al., 2013*).

## Statistics

Experimental flies and genetic controls were tested under the same conditions, and data were collected from at least two independent experiments. Statistical analysis was performed using GraphPad Prism 9 (https://www.graphpad.com/scientific-software/prism/) as indicated in each figure legend. Data were first verified for normal distribution by the D'Agostino–Pearson normality test. For normally distributed data, Student's $t$ test was used for pairwise comparisons, and one-way ANOVA was used for comparisons among multiple groups, followed by Tukey's multiple comparisons. For data not normally distributed, the Mann-Whitney test was used for pairwise comparisons, and the Kruskal–Wallis test was used for comparison among multiple groups, followed by Dunn's multiple comparisons. For group-matched data that were normally distributed, the Paired $t$-test signed rank test was used for pairwise comparisons, and RM one-way ANOVA was used for comparison among multiple groups, followed by RM one-way ANOVA multiple comparisons. For group-matched data that were not normally distributed, the Wilcoxon matched-pairs signed rank test was used for pairwise comparisons, and the Friedman test was used for comparison among multiple groups, followed by Dunn's multiple comparisons. For qPCR experiments, the average relative expression of three independent experiments was analyzed using the Mann-Whitney test. For one group compared with zero, we used a one-sample $t$-test. The Chi-square test was used to compare male tussling percentage between two groups.

## Acknowledgements

We thank the Bloomington *Drosophila* Stock Center, Tsinghua Fly Center, and Dr. Kaiyu Wang and Fei Wang for fly stocks. This work was supported by grants from National Key R&D Program of China (2021YFA1101300), the National Natural Science Foundation of China (32371067 to YP), Shenzhen Medical Academy of Research and Translation (B2402005 to YP), the Natural Science Foundation from Jiangsu Province (BK20231418 to QP), and the Fundamental Research Funds for the Central Universities (2242023R40054 to QP).

## Additional information

### Funding

| Funder | Grant reference number | Author |
| --- | --- | --- |
| National Natural Science Foundation of China | 32371067 | Yufeng Pan |
| National Key R&D Program of China | 2021YFA1101300 | Yufeng Pan |
| Shenzhen Medical Academy of Research and Translation | B2402005 | Yufeng Pan |
| Natural Science Foundation from Jiangsu Province | BK20231418 | Qionglin Peng |

| Funder | Grant reference number | Author |
| --- | --- | --- |
| Fundamental Research Funds for the Central Universities | 2242023R40054 | Qionglin Peng |

The funders had no role in study design, data collection and interpretation, or the decision to submit the work for publication.

## Author contributions

Can Gao, Conceptualization, Data curation, Formal analysis, Investigation, Methodology, Writing – original draft; Mingze Ma, Data curation, Formal analysis, Investigation; Jie Chen, Xiaoxiao Ji, Data curation, Investigation; Qionglin Peng, Conceptualization, Funding acquisition, Project administration, Writing – review and editing; Yufeng Pan, Conceptualization, Supervision, Funding acquisition, Writing – original draft, Project administration, Writing – review and editing

## Author ORCIDs

Can Gao ⓘ https://orcid.org/0000-0002-0529-9502
Qionglin Peng ⓘ https://orcid.org/0000-0002-2287-1363
Yufeng Pan ⓘ https://orcid.org/0000-0002-1535-9716

Reviewer #1 (Public review): https://doi.org/10.7554/eLife.104212.4.sa1
Reviewer #2 (Public review): https://doi.org/10.7554/eLife.104212.4.sa2
Author response https://doi.org/10.7554/eLife.104212.4.sa3

# Additional files

## Supplementary files

Supplementary file 1. Detailed information for fly stocks.

Supplementary file 2. Detailed comparison for territorial control and mating competition.

MDAR checklist

## Data availability

Data are included in the article and its supplementary files. Source data have been provided. Fly stocks and reagents used in this study are available from the corresponding author upon request.

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
