## [Editor Report · eLife Assessment]

The **important** paper presents a new behavioral assay for *Drosophila* aggression and demonstrates that social experience influences fighting strategies, with group-housed males favoring high-intensity but low-frequency tussling over aggressive lunging observed in isolated males. The experiments are **solid** and the conclusions are of interest to researchers studying the impact of social isolation on aggression.

---

## [Referee Report · Reviewer #1 (Public review)]

This work addresses an important question in the field of Drosophila aggression and mating. Prior social isolation is known to increase aggression in males, manifesting as increased lunging, which is suppressed by group housing (GH). However, it is also known that single housed (SH) males, despite their higher attempts to court females, are less successful. Here, Gao et al., develop a modified aggression assay to address this issue by recording aggression in Drosophila males for 2 hours, with a virgin female immobilized by burying its head in the food. They found that while SH males frequently lunge in this assay, GH males switch to higher intensity but very low frequency tussling. Constitutive neuronal silencing and activation experiments implicate cVA sensing Or67d neurons in promoting high frequency lunging, similar to earlier studies, whereas Or47b neurons promote low frequency but higher intensity tussling. Optogenetic activation revealed that three pairs of pC1SS2 neurons increase tussling. Cell-type-specific DsxM manipulations combined with morphological analysis of pC1SS2 neurons and side-by-side tussling quantification link the developmental role of DsxM to the functional output of these aggression-promoting cells. In contrast, although optogenetic activation of P1a neurons in the dark did not increase tussling, thermogenetic activation under visible light drove aggressive tussling. Using a further modified aggression assay, GH males exhibit increased tussling and maintain territorial control, which could contribute to a mating advantage over SH males, although direct measures of reproductive success are still needed

Strengths:

Through a series of clever neurogenetic and behavioral approaches, the authors implicate specific subsets of ORNs and pC1 neurons in promoting distinct forms of aggressive behavior, particularly tussling. They have devised a refined territorial control paradigm, which appears more robust than earlier assays. This new setup is relatively clutter-free and could be amenable to future automation using computer vision approaches. The updated Figure 5, which combines cell-type-specific developmental manipulation of pC1SS2 neurons with behavioral output, provides a link between developmental mechanisms and functional aggression circuits. The manuscript is generally well written, and the claims are largely supported by the data.

Weakness:

All prior concerns have been addressed in the revised manuscript. The added 'Limitations of the study' section is a welcome and important clarification. Despite these limitations, the study provides valuable insights into the neural and behavioral mechanisms of Drosophila aggression.

---

## [Referee Report · Reviewer #2 (Public review)]

Summary:

Gao et al. investigated the change of aggression strategies by the social experience and its possible biological significance by using Drosophila. Two modes of inter-male aggression in Drosophila are known: lunging, high-frequency but weak mode, and tussling, low-frequency but more vigorous mode. Previous studies have mainly focused on the lunging. In this paper, the authors developed a new behavioral experiment system for observing tussling behavior and found that tussling is enhanced by group rearing, while lunging is suppressed. They then searched for neurons involved in the generation of tussling. Although olfactory receptors named Or67d and Or65a have previously been reported to function in the control of lunging, the authors found that these neurons do not function in the execution of tussling and another olfactory receptor, Or47b, is required for tussling, as shown by the inhibition of neuronal activity and the gene knockdown experiments. Further optogenetic experiments identified a small number of central neurons pC1[SS2] that induce the tussling specifically. These neurons express doublesex (dsx), a sex-determination factor, and knockdown of dsx strongly suppresses the induction of tussling. In order to further explore the ecological significance of the aggression mode change in group-rearing, a new behavioral experiment was performed to examine the territorial control and the mating competition. And finally, the authors found that differences in the social experience (group vs. solitary rearing) are important in these biologically significant competitions. These results add a new perspective to the study of aggression behavior in Drosophila. Furthermore, this study discusses an interesting general model in which the social experience modified behavioral changes play a role in reproductive success.

Strengths:

A behavioral experiment system that allows stable observation of tussling, which could not be easily analyzed due to its low-frequency, would be very useful. The experimental setup itself is relatively simple, just addition of a female to the platform, so it should be applicable to future research. The finding about the relationship between the social experience and the aggression mode change is quite novel. Although the intensity of aggression changes with the social experience was already reported in several papers (Liu et al., 2011 etc), the fact that the behavioral mode itself changes significantly has rarely been addressed, and is extremely interesting. The identification of sensory and central neurons required for the tussling makes appropriate use of the genetic tools and the results are clear. A major strength of this study in the neurobiology is the finding that another group of neurons (Or47b-expressing olfactory neurons and pC1[SS2] neurons), distinct from the group of neurons previously thought to be involved in low-intensity aggression (i.e. lunging), function in the tussling behavior. Furthermore, the results showing that the regulation of aggression by pC1[SS2] neurons is based the function of the dsx gene will bring a new perspective to the field. Further investigation of the detailed circuit analysis is expected to elucidate the neural substrate of the conflicting between the two aggression modes. The experimental systems examining the territory control and the reproductive competition in Fig. 6 are novel and have advantages in exploring their biological significance. It is important to note that, in addition to showing the effects of age and social experience on territorial and mating behaviors, the authors suggested that an altered fighting strategy has effects with respect to these behaviors.

Weaknesses:

New experimental paradigm in Fig. 6 is quite useful, but as the authors mentioned, still the future investigations are needed to reveal a direct relationship between aggression strategies and reproductive success.

---

## [Author Response]

The following is the authors’ response to the previous reviews

**Reviewer #1 (Public review):**
This work addresses an important question in the field of Drosophila aggression and mating. Prior social isolation is known to increase aggression in males, manifesting as increased lunging, which is suppressed by group housing (GH). However, it is also known that single housed (SH) males, despite their higher attempts to court females, are less successful. Here, Gao et al., develop a modified aggression assay to address this issue by recording aggression in Drosophila males for 2 hours, with a virgin female immobilized by burying its head in the food. They found that while SH males frequently lunge in this assay, GH males switch to higher intensity but very low frequency tussling. Constitutive neuronal silencing and activation experiments implicate cVA sensing Or67d neurons in promoting high frequency lunging, similar to earlier studies, whereas Or47b neurons promote low frequency but higher intensity tussling. Optogenetic activation revealed that three pairs of pC1SS2 neurons increase tussling. Cell-type-specific DsxM manipulations combined with morphological analysis of pC1SS2 neurons and side-by-side tussling quantification link the developmental role of DsxM to the functional output of these aggression-promoting cells. In contrast, although optogenetic activation of P1a neurons in the dark did not increase tussling, thermogenetic activation under visible light drove aggressive tussling. Using a further modified aggression assay, GH males exhibit increased tussling and maintain territorial control, which could contribute to a mating advantage over SH males, although direct measures of reproductive success are still needed.Strengths:Through a series of clever neurogenetic and behavioral approaches, the authors implicate specific subsets of ORNs and pC1 neurons in promoting distinct forms of aggressive behavior, particularly tussling. They have devised a refined territorial control paradigm, which appears more robust than earlier assays using a food cup (Chen et al., 2002). This new setup is relatively clutter-free and could be amenable to future automation using computer vision approaches. The updated Figure 5, which combines cell-type-specific developmental manipulation of pC1SS2 neurons with behavioral output, provides a link between developmental mechanisms and functional aggression circuits. The manuscript is generally well written, and the claims are largely supported by the data.

Thank you for the precise summary of the manuscript and acknowledgment of the novelty and significance of the study.

Weakness:Although most concerns have been addressed, the manuscript still lacks a rigorous, objective method for quantifying lunging and tussling. Because scoring appears to have been done manually and a single lunge in a 30 fps video spans only 2-3 frames, the 0.2 s cutoff seems arbitrary, and there are no objective criteria distinguishing reciprocal lunging from tussling. Despite this, the study offers valuable insights into the neural and behavioral mechanisms of Drosophila aggression.

Thank you for this comment. The duration of each lunge was measured by analyzing the videos frame by frame—from the frame before the initiation of the lunge to the frame after its completion—resulting in an average span of 3–5 frames. Given a frame rate of 30 fps, this corresponds to approximately 0.1–0.17 seconds. We acknowledge that there are certain limitations for manually quantifying the two types of aggressive behaviors, which has now been stated in the newly added “Limitations of the Study” section in the revised manuscript.

**Reviewer #2 (Public review):**
Summary:Gao et al. investigated the change of aggression strategies by the social experience and its biological significance by using Drosophila. Two modes of inter-male aggression in Drosophila are known: lunging, high-frequency but weak mode, and tussling, low-frequency but more vigorous mode. Previous studies have mainly focused on the lunging. In this paper, the authors developed a new behavioral experiment system for observing tussling behavior and found that tussling is enhanced by group rearing, while lunging is suppressed. They then searched for neurons involved in the generation of tussling. Although olfactory receptors named Or67d and Or65a have previously been reported to function in the control of lunging, the authors found that these neurons do not function in the execution of tussling and another olfactory receptor, Or47b, is required for tussling, as shown by the inhibition of neuronal activity and the gene knockdown experiments. Further optogenetic experiments identified a small number of central neurons pC1[SS2] that induce the tussling specifically. These neurons express doublesex (dsx), a sex-determination factor, and knockdown of dsx strongly suppresses the induction of tussling. In order to further explore the ecological significance of the aggression mode change in group-rearing, a new behavioral experiment was performed to examine the territorial control and the mating competition. And finally, the authors found that differences in the social experience (group vs. solitary rearing) and the associated change in aggression strategy are important in these biologically significant competitions. These results add a new perspective to the study of aggression behavior in Drosophila. Furthermore, this study proposes an interesting general model in which the social experience modified behavioral changes play a role in reproductive success.Strengths:A behavioral experiment system that allows stable observation of tussling, which could not be easily analyzed due to its low-frequency, would be very useful. The experimental setup itself is relatively simple, just the addition of a female to the platform, so it should be applicable to future research. The finding about the relationship between the social experience and the aggression mode change is quite novel. Although the intensity of aggression changes with the social experience was already reported in several papers (Liu et al., 2011 etc), the fact that the behavioral mode itself changes significantly has rarely been addressed, and is extremely interesting. The identification of sensory and central neurons required for the tussling makes appropriate use of the genetic tools and the results are clear. A major strength of this study in neurobiology is the finding that another group of neurons (Or47b-expressing olfactory neurons and pC1[SS2] neurons), distinct from the group of neurons previously thought to be involved in low-intensity aggression (i.e. lunging), function in the tussling behavior. Furthermore, the results showing that the regulation of aggression by pC1[SS2] neurons is based on the function of the dsx gene will bring a new perspective to the field. Further investigation of the detailed circuit analysis is expected to elucidate the neural substrate of the conflict between the two aggression modes. The experimental systems examining the territory control and the reproductive competition in Fig. 6 are novel and have advantages in exploring their biological significance. It is important to note that in addition to showing the effects of age and social experience on territorial and mating behaviors, the authors experimentally demonstrated that altered fighting strategy has effects with respect to these behaviors.

Thank you for your precise summary of our study and being very positive on the novelty and significance of the study.

**Reviewer #3 (Public review):**
In this revised manuscript, Gao et al. presented a series of well-controlled behavioral data showing that tussling, a form of high-intensity fighting among male fruit flies (*Drosophila melanogaster*) is enhanced specifically among socially experienced and relatively old males. Moreover, results of behavioral assays led authors to suggest that increased tussling among socially experienced males may increase mating success. They also concluded that tussling is controlled by a class of olfactory sensory neurons and sexually dimorphic central neurons that are distinct from pathways known to control lunges, a common male-type attack behavior.A major strength of this work is that it is the first attempt to characterize behavioral function and neural circuit associated with Drosophila tussling. Many animal species use both low-intensity and high-intensity tactics to resolve conflicts. High-intensity tactics are mostly reserved for escalated fights, which are relatively rare. Because of this, tussling in the flies, like high-intensity fights in other animal species, have not been systematically investigated. Previous studies on fly aggressive behavior have often used socially isolated, relatively young flies within a short observation duration. Their discovery that (1) older (14-days old) flies tend to tussle more often than younger (2 to 7-days-old) flies, (2) group-reared flies tend to tussle more often than socially isolated flies, and (3) flies tend to tussle at later stage (mostly ~15 minutes after the onset of fighting), are the result of their creativity to look outside of conventional experimental settings. These new findings are key for quantitatively characterizing this interesting yet under-studied behavior.Newly presented data have made several conclusions convincing. Detailed descriptions of methods to quantify behaviors help understand the basis of their claims by improving transparency. However, I remain concerned about authors' persistent attempt to link the high intensity aggression to reproductive success. The authors' effort to "tone down" the link between the two phenomena remains insufficient. There are purely correlational. I reiterate this issue because the overall value of the manuscript would not change with or without this claim.

Thank you for acknowledging the novelty and significance of the study. Regarding the relationship you mentioned between high-intensity aggression and reproductive success, we further toned down the statement between them throughout the manuscript in the revised manuscript. We also modified the title to “Social Experience Shapes Fighting Strategies in Drosophila”. In addition, we now added a ‘Limitations of the Study’ section to clearly state the correlation between tussling and reproductive success.

**Reviewer #1 (Recommendations for the authors):**
If possible, mention the EM-connectome data showing the minimal interneuronal path from Or47b ORNs to pC1SS2 neurons (even if derived from the female connectome), which can strengthen the model of parallel sensory-central pathways.

Thank you for this comment. According to data from the EM connectome, connecting Or47b ORNs to pC1d neurons requires at least two intermediate neurons. An example minimal pathway is: ORN_VA1v (L) → AL-AST1 (L) → PLP245 (L) → pC1d (R). We have added this point in the Discussion section of the revised manuscript.

I'm not convinced that labeling lunges as "gentle" combat behavior works, either in the abstract or elsewhere. While lunging is indeed a lower-intensity form of aggression compared to tussling, applying anthropomorphic descriptors risks misleading readers.

Thank you for this comment. We now use “low-intensity” instead of “gentle” to describe lunging.

In Materials & Methods, please cross-check all figure-panel references after the recent re-numbering (e.g. "Figure 5A6A" etc.).

Thank you for this comment. We have thoroughly verified the figure panel references in the Materials & Methods section.

Ensure that Table S1 is clearly cited in the main text where you first describe fly genotypes.

Thank you for this comment. We have now cited Table S1 in the main text.

There are multiple grammatical errors and typos throughout the manuscript. Please correct them. Some examples are below, but this is not an exhaustive list:Line 98-102 requires rephrasing as the results are already published and not being observed by the authors.

Thank you for this comment. We have revised the manuscript to “we occasionally observed the high-intensity boxing and tussling behavior in male flies as previously reported (Chen et al., 2002; Nilsen et al., 2004), which….”

line 116- lower not 'lowed'.

Corrected.

line 942 & 945- knock-down males not 'knocking down males'.

Corrected. Thank you very much for these comments.

**Reviewer #2 (Recommendations for the authors):**
The authors have almost completely answered the major comments I have noted on the ver.1 manuscript: (1) They clearly show changes in fighting strategy in the territory control behavior experiment in Fig. 6-figure supplements. (2) A detailed description of how aggressive behavior is measured. Thus, I am convinced by this revision.

Thank you for these comments that make the manuscript a better version.

Furthermore, in Fig. 5, which examined the relationship of pC1[SS2] characteristics with the function of dsx, is a novel data and very interesting. I look forward to further developments.

Thank you. We will continue to explore this part in our future study.

However, one point still concerns me.Line 192: Although the authors describe it as "usage-dependent," the trans-Tango technique is essentially a postsynaptic cell-labeling technique. It is possible that the labeling intensity in postsynaptic cells increases from the change in expression levels of the Or47b gene due to GH. However, there is no difference in the expression level of the Or47b gene labeled by GFP between SH and GH. Therefore, we cannot conclude that the expression of the Or47b gene is increased by rearing conditions.The original paper on trans-TANGO (Talay et al., 2017) does not discuss the usage-dependency. A review of trans-synaptic labeling techniques (Ni, Front Neural Circuits. 2021) discusses that the increase in trans-TANGO signaling with aging may be related to synaptic strength, but there is no experimental evidence for this. In my opinion, the results in Figure 3-figure supplement 2 only weakly suggest that the increase in trans-TANGO signaling may be explained by an increase in synaptic strength due to group rearing.

We appreciate the reviewer’s insightful comment regarding the interpretation of the trans-Tango signal. Indeed, the original trans-Tango study (Talay et al., 2017) does not claim that the method is usage-dependent. The observed increase in trans-Tango labeling with age, as reported in their supplemental figures, may reflect accumulation over time, potentially influenced by synaptic maturation or increased component expression. To avoid overstating our results, we have revised the relevant statement in the manuscript to remove the term "usage-dependent" and now describe the change in trans-Tango signal more cautiously.

**Reviewer #3 (Recommendations for the authors):**
Below are the cases where their professed attempts to "tone down the statement" appear ignored:Lines 27-29:"Our findings... suggest how social experience shapes fighting strategies to optimize reproductive success".

We have now revised the manuscript to “Our findings… suggest that social experience may shape fighting strategies to optimize reproductive success.”

Lines 85-86:"... discover that this infrequent yet intense form of combat is... crucial for territory dominance and mating competition".

We have now revised the manuscript to “…discover that this infrequent yet intense form of combat is enhanced by social enrichment, while the low-intensity lunging is suppressed by social enrichment.”

Lines 335-339:"Here, we found that... GH males tend to... increase the high-intensity tussling, which enhances their territorial and mating competition."

We have removed “which enhances their territorial and mating competition” in the revised manuscript.

Lines 343-344:"... presenting a paradox between social experience, aggression and reproductive success. Our result resolved this paradox..."

We have now revised the manuscript to “...Our results provide an explanation for this paradox…”

Lines 355-358:"Interestingly, we found that the mating advantage gained through social enrichment can even offset the mating disadvantage associated with aging, further supporting the vital role of shifting fighting strategies in experienced, aged males."

We have removed “further supporting the vital role of shifting fighting strategies in experienced, aged males” in the revised manuscript.

Lines 361-362:"These results separate the function of the two fighting forms and rectify out understanding of how social experiences regulate aggression and reproductive success."

We have removed this sentence in the revised manuscript.

Some may say that a speculative statement is harmless, but I think it indeed is harmful unless it is clearly indicated as a speculation. It is regrettable that authors remain reluctant to change their claim without providing any new supporting evidence. All three reviewers raised the same concern in the first round of review.

We apologize for not making the speculative nature of the statement clearer in the previous version. In the revised manuscript, we have now explicitly rephrased sentences to only suggest a correlation but not a causal link between tussling and reproductive success.

I have no choice but to keep my evaluation of the manuscript as "Incomplete" unless the authors thoroughly eliminate any attempt to link these two. This must go beyond changing a few words in the lines listed above.

Thank you for this comment. In addition to the lines listed above, we carefully checked all statements regarding the correlation between fighting strategies and reproductive success throughout the full text. Furthermore, we have also added a “Limitations of the Study” section to address the shortcomings of this study in the revised manuscript.

I do not have the same level of concern over the interpretation of Fig. 6A-C, because this is directly linked to aggressive interactions. Even if the socially isolated males do not engage in tussling, it is not a leap to assume that a different fighting tactic of socially experienced males can give them an advantage in defending a territory. To me, this is a sufficient ethological link with the observed behavioral change.

Thank you for this insightful comment.

The following are relatively minor, although important, concerns.I beg to differ over the authors' definition of "tussling". Supplemental movies S1 and S2 appear to include "tussling" bouts in which 2 flies lunging at each other in rapid succession, and supplemental movie S3 appears to include bouts of "holding", in which one fly holds the opponent's wings and shakes vigorously. These cases suggest that the definition of "tussling" as opposed to "lunging" has a subjective element. However, I would not delve on this matter further because it is impossible to be completely objective over behavioral classification, even by using a computational method. An important point is that the definition is applied consistently within the publication. I have no reason to doubt that this was not the case.

Thank you for this comment. Since the analysis of tussling behavior was conducted manually, it is challenging to achieve complete objectivity. However, we made every effort to apply consistent criteria throughout the analysis. We have added a “Limitations of the Study” section in the revised manuscript to clearly state this caveat. We appreciate your understanding.

Authors now state that "all tester flies were loaded by cold anesthesia" (lines 432-433). I would like to draw attention to the well-known fact that anesthesia, whether by ice or by CO2, are long known to affect fly's subsequent behaviors (for aggression, see Trannoy S. et al., Learn. Mem. 2015. 22: 64-68). It will be prudent to acknowledge the possibility that this handling method could have contributed to unusually high levels of spontaneous tussling, which has not been reported elsewhere before.

Thank you for this comment. The increased tussling behavior observed in our study is unlikely due to cold anesthesia, as noted by Trannoy S. et al. (2015), cold anesthesia profoundly reduces locomotion and general aggressiveness in flies. We acknowledge that the use of cold anesthesia in behavioral experiments may have potential effects on aggression. To minimize this influence, we allowed the flies to recover and adapt for at least 30 minutes before behavioral recording. Moreover, both control and experimental groups were treated in exactly the same manner to ensure consistency.

It is intriguing that pC1SS2 neurons are dsx+ but fru-. Authors convincingly demonstrated that these neurons are clearly distinct from the P1a neurons, a well-characterized hub for male social behaviors. It is possible that pC1SS2 neurons overlap with previously characterized dsx+ neurons that are important for male aggressions (measured by lunges), such as in Koganezawa et al., Curr. Biol. 2016 and Chiu et al., Cell 2020, a point authors could have explicitly raised.

Thank you for this comment. We have added this point into the Discussion section of the revised manuscript, as follows: “That tussling-promoting… aggression (Koganezawa et al., 2016). Moreover, the anatomical features of pC1^SS2^ neurons are highly similar to the male-specific aggression-promoting (MAP) neurons identified by another previous study (Chiu et al., 2021).

I acknowledge the authors' courage to initiate an investigation to a less characterized, high intensity fighting behavior. Tussling requires the simultaneous engagement of two flies. Even if there are confusion over the distinction between lunges and tussling, authors' conclusion that socially experienced flies and socially isolated flies employ distinct fighting strategy is convincing. The concern I raised above is about the interpretation of the data, not about the quality of data.

Thank you for your constructive comments to make this manuscript better.